# Interactions between Dpr11 and DIP-γ control selection of amacrine neurons in *Drosophila* color vision circuits

**Kaushiki P Menon[1]\*, Vivek Kulkarni[1], Shin-ya Takemura[2], Michael Anaya[1], Kai Zinn[1]\***

[1]Division of Biology and Biological Engineering, California Institute of Technology, Pasadena, United States; [2]Janelia Research Campus, Howard Hughes Medical Institute, Ashburn, United States

**Abstract** *Drosophila* R7 UV photoreceptors (PRs) are divided into yellow (y) and pale (p) subtypes. yR7 PRs express the Dpr11 cell surface protein and are presynaptic to Dm8 amacrine neurons (yDm8) that express Dpr11's binding partner DIP-γ, while pR7 PRs synapse onto DIP-γ-negative pDm8. Dpr11 and DIP-γ expression patterns define 'yellow' and 'pale' color vision circuits. We examined Dm8 neurons in these circuits by electron microscopic reconstruction and expansion microscopy. *DIP-γ* and *dpr11* mutations affect the morphologies of yDm8 distal ('home column') dendrites. yDm8 neurons are generated in excess during development and compete for presynaptic yR7 PRs, and interactions between Dpr11 and DIP-γ are required for yDm8 survival. These interactions also allow yDm8 neurons to select yR7 PRs as their appropriate home column partners. yDm8 and pDm8 neurons do not normally compete for survival signals or R7 partners, but can be forced to do so by manipulation of R7 subtype fate.

**\*For correspondence:**
menonk@caltech.edu (KPM);
zinnk@caltech.edu (KZ)

**Competing interests:** The authors declare that no competing interests exist.

## Introduction

The chemoaffinity hypothesis (*Sperry, 1963*) proposed that assembly of neural circuits involves interactions among cell-surface proteins (CSPs) that label individual neurons or neuronal types. This hypothesis still applies well to systems such as the *Drosophila* brain and vertebrate retina, in which synaptic connectivity is largely controlled by intrinsic gene expression patterns. Identifying synaptic target labels is a major challenge, however, because individual neurons can express hundreds of different CSPs. The *Drosophila* genome encodes about 1000 CSPs involved in cell-cell interactions (*Kurusu et al., 2008*). A study of seven neuronal types in the pupal visual system found that each type expresses more than 250 of these CSP genes and differs from any of the other types by expression of least 50 genes (*Tan et al., 2015*).

CSPs involved in neural circuit assembly include heterophilic interaction partners expressed on synaptically connected neurons. Among these are Dprs and DIPs, which are members of an interacting network of immunoglobulin superfamily (IgSF) CSPs called the 'Dpr-ome' (for review see *Zinn and Özkan, 2017*). The Dpr-ome was discovered in an in vitro 'interactome' screen of all *Drosophila* IgSF proteins (*Carrillo et al., 2015*; *Özkan et al., 2013*). It has 21 2-IgSF domain Dpr proteins (*Nakamura et al., 2002*), each of which binds to one or more of the 11 3-IgSF DIP proteins. Most DIPs interact with multiple Dprs and vice versa, and their binding affinities vary between 1 and 200 μM (*Cheng et al., 2019a*; *Cosmanescu et al., 2018*). Each DIP and Dpr is expressed in a unique subset of neurons in embryos, larvae, and pupae. In the pupal visual system, neurons expressing a DIP tend to be postsynaptic to neurons that express a Dpr to which that DIP binds in vitro (*Carrillo et al., 2015*; *Cosmanescu et al., 2018*; *Davis et al., 2018*; *Tan et al., 2015*; *Xu et al., 2018*). DIPs and Dprs define an IgSF family, present in both protostomes and deuterostomes, that

has been denoted as the Wirins. The five mammalian IgLON proteins are members of the Wirin family (*Cheng et al., 2019b*).

DIP-Dpr interactions can have a variety of functions. DIP-γ and Dpr11 regulate neuromuscular junction (NMJ) morphology in larvae, and control survival of DIP-γ-expressing postsynaptic cells in the pupal optic lobe (OL) (*Carrillo et al., 2015*; *Xu et al., 2018*). Interactions between postsynaptic DIP-α and presynaptic Dprs 6 and 10 also control survival of postsynaptic OL neurons, and can determine their synaptic connectivity patterns (*Xu et al., 2018*). In the larval and adult neuromuscular systems, however, DIP-α and Dpr10 control branching of DIP-α-expressing motor axons onto muscle fibers that express Dpr10 (*Ashley et al., 2019*; *Cheng et al., 2019a*; *Venkatasubramanian et al., 2019*). Dprs and DIPs regulate fasciculation and sorting of olfactory receptor neuron axons in the antennal lobe (*Barish et al., 2018*). In the lamina of the OL, DIPs prevent ectopic synapse formation (*Xu et al., 2019*).

These data are consistent with the idea that DIPs and Dprs function as synaptic target labels in certain contexts, but can also have other roles. Each neuron may acquire a unique 'surface identity' through its expression of a large collection of CSPs, among which are DIPs and Dprs. Each neuron uses its unique repertoire of CSPs to sculpt its morphology, determine its synaptic connectivity, and regulate its physiological properties. The total number of CSP genes is only about fourfold larger than the size of a repertoire (*Tan et al., 2015*), so the repertoires of different neurons necessarily overlap. Two neuronal types might have some of the same CSPs in their repertoires but use them in different ways, depending on their developmental and contextual needs.

In this paper, we examine the functions of the DIP-γ-Dpr11 interaction pair in the development of color vision circuits in the OL. The Drosophila compound eye is composed of ~750 ommatidial units. Each ommatidium contains eight photoreceptors (PRs) that express different rhodopsins (Rh). Outer PRs (R1-R6) are used for motion detection, while the R7 and R8 inner PRs transmit chromatic information (*Figure 1A–B*). R7 is a UV receptor, while R8 responds to visible light. Visual information received in the retina is transmitted to the OL, which consists of the lamina, medulla and lobula complex (*Figure 1C*). The lamina and medulla are organized retinotopically, so that each point in visual space corresponds to one cartridge in the lamina and one column in the medulla.

R1-R6 synapse onto lamina neurons (L1-L5), which in turn project to layers M1-M5 of the medulla. R7 and R8 axons grow through the lamina and into the medulla, where they terminate in layers M6 (R7) and M3 (R8) (*Figure 1C*). The medulla contains about 100 types of neurons. Some of these arborize only in the medulla, either in single columns or across multiple columns, while others have dendrites in the medulla and project to higher order visual areas, including the lobula and lobula plate (for reviews see *Kolodkin and Hiesinger, 2017*; *Millard and Pecot, 2018*; *Plazaola-Sasieta et al., 2017*). One of the major synaptic targets of R7 PRs in medulla layer M6 is Dm8, a wide-field amacrine neuron that pools information from 12 to 16 R7s (*Gao et al., 2008*).

There are two major types of ommatidia, pale (p) and yellow (y), which are randomly distributed in the retina. R7 and R8 PRs in these ommatidia are divided into subtypes with different spectral sensitivities. p ommatidia (~35%) detect shorter wavelengths, and have R7 that express the Rh3 (shorter-wave UV) rhodopsin and R8 that express Rh5 (blue), while y ommatidia (~65%) detect longer wavelengths, and have R7 that express Rh4 (longer-wave UV) and R8 that express Rh6 (green). The R7 and R8 within an ommatidium mutually inhibit each other (*Schnaitmann et al., 2018*). A third type of ommatidia, 'DRA' is found in two rows in the dorsal rim area of the eye and is important for polarized light detection. The R7 and R8 in these ommatidia both express Rh3 (reviewed by *Viets et al., 2016*; *Figure 1B*).

*Drosophila* phototaxes to UV (R7) in preference to visible light (R8). It also exhibits true color vision, being able to make intensity-independent discriminations among hues that differentially stimulate p and y R8 channels (*Melnattur et al., 2014*; reviewed by *Song and Lee, 2018*). To be able to distinguish blue from green, or short from long-wavelength UV, the fly must have different neural responses to stimulation of p and y R8 and R7 channels and utilize these response profiles to control its actions.

In our earlier work, we showed that Dpr11 is selectively expressed by yR7 PRs, while its binding partner DIP-γ is expressed by a subset of Dm8 amacrine neurons (*Carrillo et al., 2015*). Here, we show that yR7 PRs specifically connect to DIP-γ-expressing Dm8 neurons (yDm8), while pR7 PRs connect to DIP-γ-negative Dm8 neurons (pDm8) in their respective y and p 'home columns'. Analysis of the electron microscopic (EM) reconstruction of the medulla (*Takemura et al.,*

2013; *Takemura et al., 2015*) in light of this connection pattern shows that there are separate 'yellow' and 'pale' circuits that could be used for discriminating long and short-wavelength UV inputs. The yellow circuit might be constructed using DIP-γ-Dpr11 interactions, since both yDm8 and Tm5a projection neurons, which are also selectively connected to yR7 PRs, express DIP-γ (*Cosmanescu et al., 2018*; *Karuppudurai et al., 2014*). DIP-γ and Dpr11 are both required for normal morphogenesis of yDm8 distal dendrites, which fasciculate with yR7 terminals and contain many of the R7-Dm8 synapses.

Given the existence of separate yellow and pale circuits, how does the system ensure that each yR7 has a yDm8 partner? Our results show that DIP-γ-expressing yDm8 neurons are generated in excess during development and compete for presynaptic yR7 partners. yR7 PRs and yDm8 neurons recognize each other using Dpr11- DIP-γ interactions. The engagement of DIP-γ by Dpr11 is necessary for generation of signals that allow yDm8 neurons to survive.

## Results

### Dm8 and R7 subtypes are present in matching ratios

DIP-γ interacts with four Dpr partners with similar affinities: Dprs 11, 15, 16 and 17 (*Figure 1D*). A subset of R7 PRs in pupal retina express GFP from the $dpr11^{MiMIC}$ reporter line (henceforth denoted as $dpr11^{Mi>GFP}$). These were confirmed to be yR7 by co-labeling with a Rhodopsin marker, Rh4-LacZ, in late pupae (*Figure 1B, E–F*). Reporters for Dprs 15, 16, and 17 are not detectably expressed in the pupal retina.

Here, we classify two subtypes of Dm8 neurons, the major synaptic partners of R7 PRs: those which express DIP-γ (yDm8) and those which do not (pDm8) (*Figure 1G–G'*). We determined the yDm8 population in the adult medullary cortex as those cells that express RFP under the control of a late pupal pan-Dm8 driver, R24F06-GAL4 (*Nern et al., 2015*) and GFP from the $DIP-γ^{MiMIC}$ reporter line (henceforth denoted as $DIP-γ^{Mi>GFP}$). The pDm8 population was identified as those cells that express RFP but not GFP; there are no known markers or drivers that selectively label pDm8 neurons. We additionally used the transcription factor Dachshund (Dac), which is expressed in both yDm8 and pDm8 neurons (*Hasegawa et al., 2011*), together with $DIP-γ^{Mi>GFP}$ to selectively identify yDm8 cell bodies in early pupal stages and to confirm yDm8 identity in adults independently of the driver. yR7 and pR7 ommatidia are present at a ~65y:35p ratio, and because of retinotopy we expect to find a similar ratio of y and p columns in the medulla. We find that yDm8 and pDm8 neurons in wild-type are present at a ratio of 60y:40p, which is comparable to the ratio of the input R7 PRs (*Figure 1B,H–I*) (*Viets et al., 2016*).

Dm8 neurons are both unicolumnar and multicolumnar in their coverage of columns in the neuropil. The arbor of each Dm8 contacts 12–16 medulla columns, but most synapses are made with the R7 in the central (home) column (*Fischbach and Dittrich, 1989*; *Gao et al., 2008*; *Takemura et al., 2013*; *Takemura et al., 2015*). *Figure 2A* shows a horizontal view (side-view as in schematic in *Figure 1C*) of a rendering of an R7 terminal and a Dm8 arbor from an EM reconstruction (*Takemura et al., 2015*). The thick bundle of dendritic processes at the center of the arbor makes extensive contacts with the R7 home column, and contains the majority of R7-Dm8 synapses (*Figures 2A* and *3*). These central dendritic projections extend distally from M6 to M4, and we have denoted them as the 'sprig' of a Dm8 arbor. The lateral arbor of each Dm8 overlaps extensively with other Dm8 arbors, but the center of highest arbor density, the sprig, tiles the medulla. The coverage pattern of a typical Dm8 is thus indicative of approximately one cell per column.

Although the ratios of the input PRs and output targets match, the Dm8 cell numbers we obtained (266 yDm8 and 179 pDm8) are insufficient for innervation of the expected 730–785 yR7 plus pR7 columns in the medulla (*Posnien et al., 2012*) even when we account for the fact that some Dm8 neurons (~6%; see below) have two home columns. This suggests that the Dm8 driver is not completely penetrant. Indeed, when we use Dac and $DIP-γ^{Mi>GFP}$ to determine yDm8 populations, we obtain a larger number (~320), although this is still less than the expected total number of yDm8 neurons, which is 440–480.

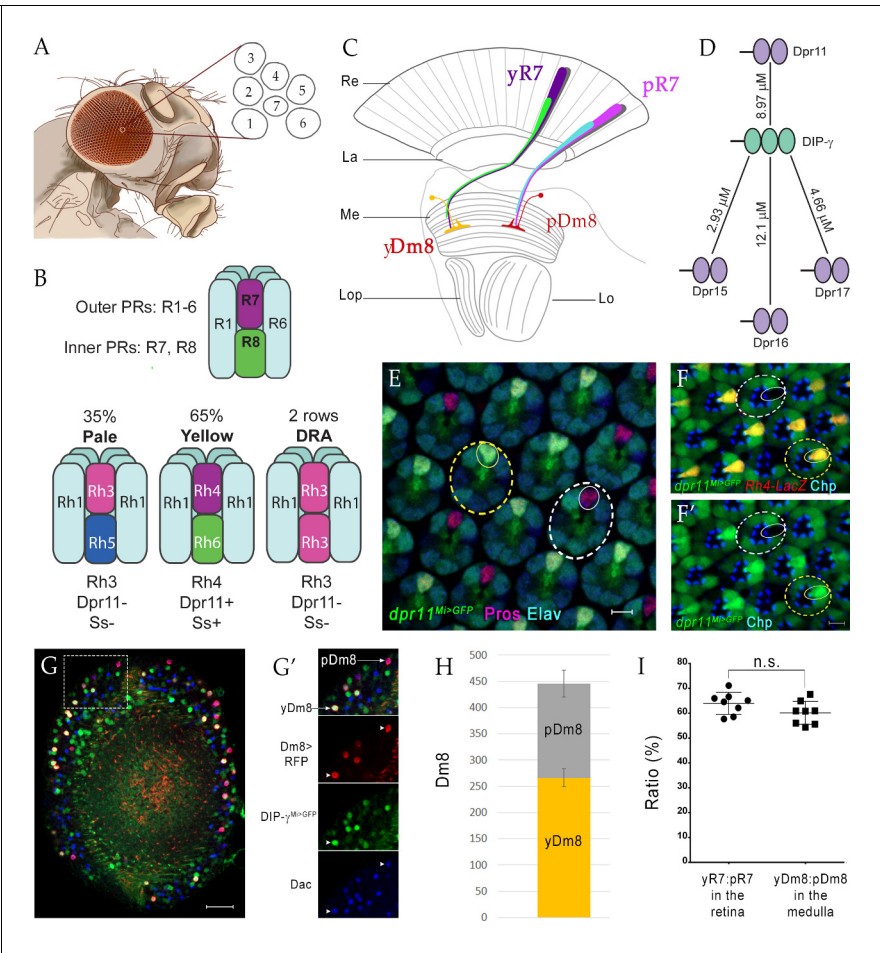

**Figure 1.** yDm8 and pDm8 populations are present in the same ratio as their presynaptic input yR7 and pR7 photoreceptors. (**A–C**) Overview of the Drosophila visual system. (**A**) Compound eye. Each ommatidium contains 8 PRs. Rhabdomeres of 7 PRs are shown in diagram, because R7 is stacked on top of R8. (**B**) R1-6 are outer PRs, and R7 and R8 are inner PRs. Three ommatidial subtypes in the retina: yellow (y), pale (p) and Dorsal Rim Area (DRA). Each ommatidium is assigned as y or p based on Rh expression patterns in R7 and R8 PRs. y and p ommatidia are distributed randomly in a ~65y:35p ratio in wild-type. Rh4, Dpr11 and Spineless transcription factor (Ss) are expressed in R7 in y ommatidia only. (**C**) Schematic of the adult optic lobe. yR7 and pR7, and yR8 (green) and pR8 (cyan) project to M6 and M3 layers of the medulla, respectively. The axons of outer PRs (gray) terminate in the lamina. yR7 and pR7 synapse on yDm8 and pDm8 in M6. Re: Retina; La: Lamina; Me: Medulla; Lop: Lobula plate; Lo: Lobula. (**D**) The DIP-γ hub consists of Dprs 11, 15, 16 and 17. Dprs are 2-IgSF domain CSPs that interact with DIPs, which are 3-IgSF domain CSPs. $K_D$s shown here are from **Cosmanescu et al. (2018)**. Ig domains indicated by ovals. (**E**) $dpr11^{Mi>GFP}$ is expressed in R7 (small yellow circle) in select ommatidia (yellow dashed circle) and absent in others (white dashed circle) in the retina. R7 indicated by small circles. Mid-pupal retina labeled with anti-Pros for all R7 PRs (magenta), anti-GFP for $dpr11^{Mi>GFP}$ reporter (green) and anti-Elav for all neurons (blue). Maximum intensity projection; scale bar 5 µm. (**F–F'**) yR7 co-expresses $dpr11^{Mi>GFP}$ and Rh4-LacZ. Dpr11 is expressed in Rh4[+] R7 in yellow ommatidia (yellow dashed circle) and absent in Rh3[+] R7 in pale ommatidia (white dashed circle). Late pupal retina labeled with anti-β-galactosidase for Rh4-LacZ reporter (red), anti-GFP for $dpr11^{Mi>GFP}$ reporter (green), and anti-Chaoptin (Chp; blue) for all PRs. Single confocal slice; scale bar 5 µm. (**G–G'**) yDm8 and pDm8 cell bodies in adult medullary cortex. Adult optic lobes labeled with anti-RFP for pan-Dm8 driver >RFP (red), anti-GFP for $DIP$-$γ^{Mi>GFP}$ reporter (green) and anti-Dac for transcription factor Dachshund (blue). yDm8 expresses RFP, GFP and Dac and pDm8 expresses RFP and Dac. Inset in G shown in G'. yDm8 and pDm8 cell bodies indicated (arrows in merged, arrowheads in individual panels). Maximum intensity projection; scale bar 20 µm. (**H**) yDm8 and pDm8 populations in adult medullary cortex. The cell numbers of y and p Dm8 neurons determined with the pan-Dm8 driver and $DIP$-$γ^{Mi>GFP}$ are indicated on the y-axis. yDm8: 266.4+ /- 17.2, pDm8: 179.4+ /- 25.6 (n = 8–11 OLs; error bars indicate std. deviation). (**I**) Matching of yDm8:pDm8 ratio in the medulla with yR7:pR7 ratio in the retina. R7 and Dm8 ratios were determined as in F and G-H, respectively. Ratios (expressed as percentages) are shown on the y-axis. Graph shows mean + /- std. deviation and unpaired Student's t-test p-values. yR7:pR7 63.9+ /- 4.4 (n = 8 retinas), yDm8:pDm8 60.2+ /- 4.6 (n = 8 OLs), not significant (n.s.). Complete genotypes in **Table 2** of Materials and methods.

## yDm8 selectively innervate yR7 in their home columns and avoid pR7

To evaluate the specificity of connections between Dm8 and R7 subtypes, we generated single-cell Dm8 (flipout) clones. For yDm8 flipouts, we used an intersectional strategy, since *DIP-γ* is expressed in many non-Dm8 cells. This employed a split-GAL4 driver that includes *DIP-γ^{MiMIC} GAL4-DBD* and *R24F06 p65-AD* hemi-drivers to selectively label DIP-γ-expressing yDm8 neurons (*Figure 2B,D*, *Figure 2—figure supplement 1C–C'*). For Dm8 that do not express DIP-γ (pDm8) flipouts, we used the pan-Dm8 driver and *Rh4-lacZ* reporter to identify clones whose home column R7 is labeled by the pan-PR marker Chp but lacks LacZ labeling, since there is no pDm8-specific driver (*Figure 2C,E*). yDm8 and pDm8 arbors have similar morphologies, with the prominent dendritic sprig identifying the home column (*Figure 2A–C*) (*Gao et al., 2008*; *Nern et al., 2015*). Using the yDm8 split-GAL4 driver, we exclusively labeled Dm8 neurons that have yR7 as the home column (43/43 flipouts; *Figure 2B,D*; *Figure 2—figure supplement 1A*). This indicates that yDm8 specifically innervate yR7 and avoid pR7 in the home column. We also observed yDm8 with two home columns at a frequency of ~6% (4/64 clones) (*Figure 2—figure supplement 1A*). All four had both sprigs on yR7 columns (see Figure 5H for an example). Since we find that all DIP-γ-positive Dm8 neurons have yR7 home columns, we can infer that cells labeled using the pan-Dm8 driver that have pR7 home columns must be the DIP-γ-negative pDm8 subtype. Single-cell clones of pDm8 were found to innervate only pR7 (21 pDm8 clones were analyzed; one had two pR7 home columns) (*Figure 2C,E*, *Figure 2—figure supplement 1A*). Specificity was further confirmed by labeling mid-pupal optic lobes with DIP-γ antibody and *dpr11^{Mi>GFP}*. DIP-γ labeling is seen in the M6 layer only under yR7 that co-express Dpr11 and Chp, and not under pR7 that are labeled with Chp alone (*Figure 2F*).

We analyzed neighboring R7 contacted by yDm8 and pDm8 in single-cell clones. Outside of their home columns, yDm8 and pDm8 dendritic arbors contact both types of R7. Cross-sectional (top-down) views show that although there is no significant difference in the total number of R7 terminals contacted by the two Dm8 subtypes, there is a bias in the type of R7 they contact (*Figure 2D–E,I–J*). yDm8 contacts with yR7 columns were significantly higher (~15%) than pDm8 connections with yR7.

The two subtypes of Dm8 are distinguished by the expression of DIP-γ. To evaluate whether the yDm8 and pDm8 populations arise independently or originate as one population, with DIP-γ being switched off in pDm8 neurons at a later time, we used *DIP-γ GAL4* to express FLP in a line with a flipout cassette (LexAop-FRT-stop-FRT) driving a GFP reporter and a pan-Dm8 LexA driver and examined GFP labeling in the M6 layer (*Figure 2G*). If yDm8 and pDm8 originate as separate populations, we would expect gaps to be present in M6 (like those seen with *DIP-γ^{Mi>GFP}* (*Figure 2H'*; schematic in *Figure 2—figure supplement 1B*) and not a continuous line in M6 (like that seen when both yDm8 and pDm8 are labeled with a pan-Dm8 driver (*Figure 2H''*)). This is because the conditional *DIP-γ* GFP reporter would be expressed only in yDm8 and absent in pDm8. The gaps represent 'pale' columns with a pR7 and have a pDm8 arbor, which does not label with *DIP-γ^{Mi>GFP}* (or with DIP-γ antibody; *Figure 2F*). *Figure 2G* shows that there are gaps in the M6 layer, indicating that the two Dm8 subtypes are in fact separate populations.

## Yellow and pale-specific synaptic connections in color vision circuits

Most R7 output synapses are made onto Dm8, Dm9 (a multicolumnar medulla intrinsic neuron), Tm5a, Tm5b, and Tm5c (*Gao et al., 2008*; *Karuppudurai et al., 2014*; *Takemura et al., 2013*; *Takemura et al., 2015*). Tm5a/b/c neurons project to the lobula, and are likely to be the main output neurons of R7 circuits. To define the synaptic connections of these neurons and determine whether they differ between y and p columns, we examined the EM reconstruction of the medulla (*Takemura et al., 2013*; *Takemura et al., 2015*). This reconstruction attempted to identify all synaptic connections between the 'home' column in the center and the six columns surrounding it (columns A-F in *Figure 3* diagram). The surrounding columns (G-R) were also partially annotated, but these columns vary in the extent to which their synapses can be assigned to specific cells. There are eight columns for which R7, Dm8, and Tm5a/b synapses are annotated (home, A, B, D, E, F, J, and K), and analysis of these columns allowed us to derive the conclusions described below.

The morphologies of y and p R7 and Dm8 neurons, as visualized in the EM reconstruction, do not allow us to distinguish their subtypes. However, the main dendritic branch of Tm5a was found to associate with yR7 but not with pR7 axons in a light-level analysis (*Karuppudurai et al., 2014*), suggesting that yR7 selectively synapses onto Tm5a. Using this hypothesis as a guide, we identified five

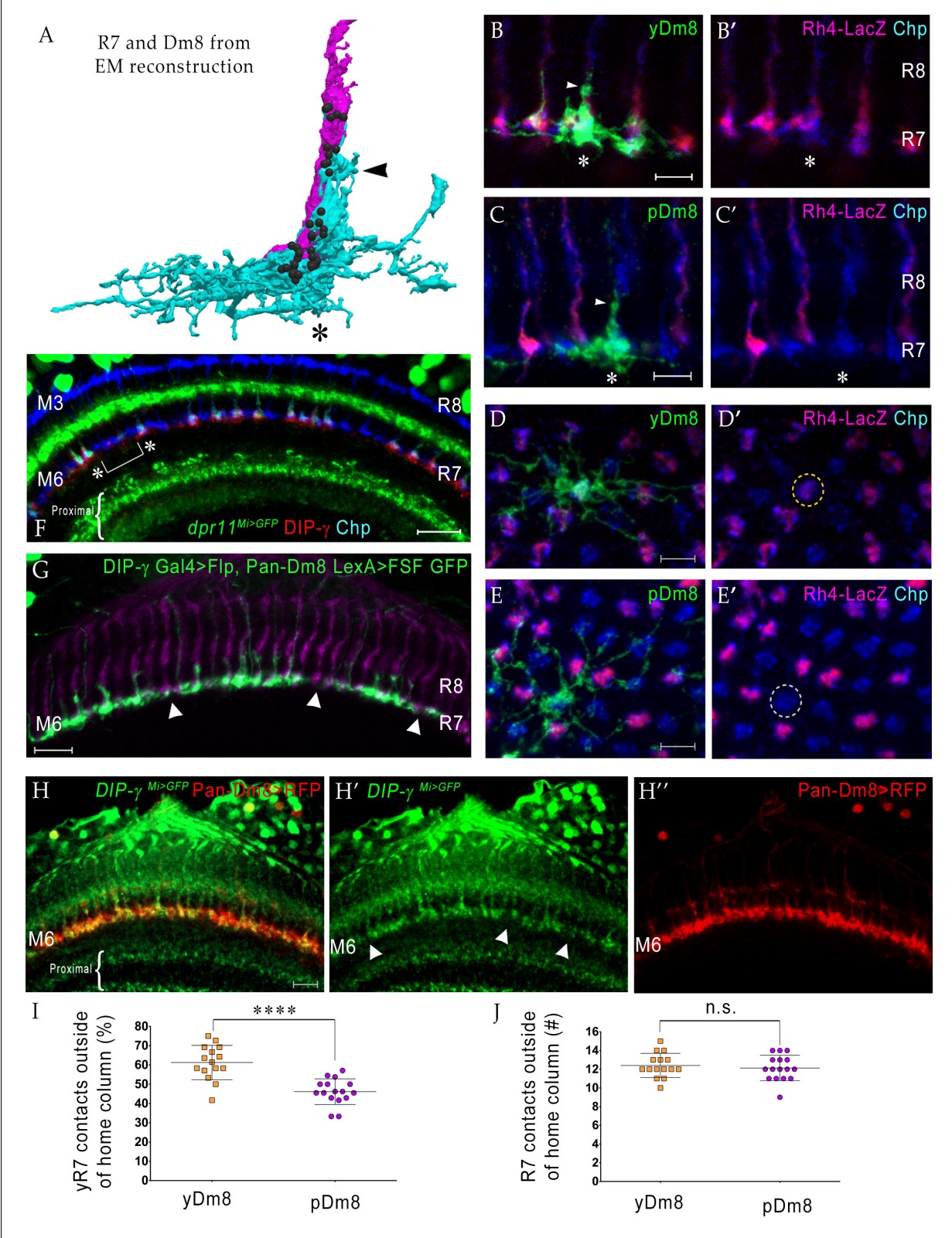

**Figure 2.** yDm8 neurons selectively innervate yR7 in their home columns and avoid pR7, and vice versa. (**A**) Rendering of a yR7 terminal and a yDm8 arbor from an EM reconstruction, in a horizontal view. The distal dendritic projection (sprig; arrowhead) of Dm8 extensively contacts the R7 home column (asterisk). Dm8, cyan; R7, magenta. Black balls indicate R7 T-bars (output synapses). (**B–E**) Single flipout clones generated either with the *DIP-γ* split-Gal4 Dm8 driver (denoted as yDm8 split-Gal4 driver in *Figure 2—figure supplement 1A*) for yDm8 or with pan-Dm8 driver for pDm8. Adult optic

*Figure 2 continued on next page*

Figure 2 continued

lobes labeled with anti-GFP for flipout clone, anti-LacZ for yR7 reporter *Rh4-lacZ* and anti-Chp for all PRs. Medulla columns were identified as pale or yellow with yR7 reporter (magenta) and Chp (blue); pR7 columns were identified by the absence of Rh4-LacZ labeling. Maximum intensity projection; scale bar 5 µm. (B–B') Horizontal view of a yDm8. The yDm8 distal dendritic projection (sprig; arrowhead) extends distally along the home column yR7 (asterisk) to the M4 layer. (C–C') Horizontal view of a pDm8. pDm8 has a similar morphology to yDm8, with the sprig (arrowhead) in contact with a pR7 home column (asterisk). (D–D') Cross-sectional (top-down) view of a yDm8. The dendritic arbor of this yDm8 contacts 13 columns (8y and 5 p). Home column yR7 indicated by yellow dashed circle. (E–E') Cross-sectional view of a pDm8. The dendritic arbor of this pDm8 contacts 14 columns (8 p and 6y). Home column pR7 indicated by white dashed circle. (F) DIP-γ-expressing yDm8 neurons specifically contact Dpr11-expressing yR7 home columns. Horizontal view of mid-pupal medulla labeled with anti-DIP-γ (red), anti-GFP for *dpr11^{Mi>GFP}* reporter (green) and anti-Chp for all PRs (blue). All yR7 PRs have DIP-γ labeling abutting the R7 (asterisks) and none of the pR7 PRs have any DIP-γ labeling apposed to them (bracket). Maximum intensity projection; scale bar 10 µm. (G) yDm8 and pDm8 populations have independent origins. The dendritic arbors of yDm8 neurons are labeled in flies carrying DIP-γ Gal4 >Flp and pan-Dm8 LexA >LexAop FRT-stop-FRT GFP transgenes. pDm8 that are not labeled appear as gaps (arrowheads) in the M6 layer, similar to the pattern of M6 layer in *DIP-γ^{Mi>GFP}* where only yDm8 express DIP-γ (H'). Adult optic lobes were labeled with anti-GFP (green) and R7 and R8 PRs were labeled with anti-Chaoptin (magenta). Maximum intensity projection; scale bar 10 µm. (H–H") Gaps representing pDm8 arbors are present in M6 layers labeled with the *DIP-γ^{Mi>GFP}* reporter (H'), but not in M6 labeled with the pan-Dm8 reporter, which labels both Dm8 subtypes (H"). Adult optic lobes labeled with anti-GFP for *DIP-γ^{Mi>GFP}* reporter (H') and anti-RFP for pan-Dm8 Gal4 >RFP (H"). Gaps are marked (arrowhead in H') in the M6 layer. The pan-Dm8 driver also labels lamina neuron L3, which is seen as faint labeling above the Dm8 layer. Maximum intensity projection; scale bar 10 µm. (I) yDm8 have a bias for contacting yR7 outside of the home column, while pDm8 contact yR7 and pR7 equally. Single flipout clones were labeled as in *Figure 2D–E* and analyzed with Imaris software to analyze connections with neighboring columns. yR7 contacts made outside of the home column: yDm8 61.3+ /- 8.9 (n = 15), pDm8 46.2+ /- 6.7 (n = 16), ****p<0.0001 (J) The total number of R7 contacted outside of the home column by the two Dm8 subtypes are the same. Single flipout clones were labeled as in *Figure 2D–E* and analyzed with Imaris software to analyze connections with neighboring columns. Total R7 contacted: yDm8 12.4+ /- 1.3 (n = 15), pDm8 12.1+ /- 1.4 (n = 16), not significant (n. s.). Complete genotypes in *Table 2* of Materials and methods.

The online version of this article includes the following figure supplement(s) for figure 2:

**Figure supplement 1.** Quantitation of yDm8 and pDm8 flipout clones.

candidate yR7 PRs (in columns A, E, F, J, and K) and three pR7 PRs (in columns Home, B and D) (*Figure 3*; y/p column identity indicated in diagram). There are seven Dm8 neurons whose home columns correspond to these eight R7 PRs (Dm8-B/home has two home columns). Each R7 makes many more output synapses onto its home column Dm8 than onto other Dm8 neurons, allowing us to identify B/home and D as pDm8, while A, E, F, J, and K are yDm8 (*Figure 3*).

To analyze projection neuron input specificity, we counted synapses made by y and p R7 PRs onto Tm5a and Tm5b. The five yR7 make output synapses only onto Tm5a and not Tm5b, which was the basis for their assignment as y (*Karuppudurai et al., 2014*). The three pR7 synapse onto Tm5b and not Tm5a (*Figure 3*). Thus, Tm5a and Tm5b represent separate output channels for y (Rh4) and p (Rh3) inputs for this set of columns (see *Figure 3—source data 1* legend). Tm5a expresses DIP-γ, while Tm5b does not (*Cosmanescu et al., 2018*). This suggests that the selective connection of yR7 to both yDm8 and Tm5a in the home column might involve Dpr11-DIP-γ interactions. There are no GAL4 drivers that distinguish between Tm5a and Tm5b, so we cannot determine if yR7-Tm5a connections are altered in mutants.

Dm8 arbors contain both pre- and postsynaptic elements, and Dm8 is presynaptic to Tm5a/b and Dm9. We find that Dm8 output synapses onto Tm5a/b have a similar specificity to the R7 output synapses. Both pDm8 neurons almost exclusively synapse onto Tm5b and not Tm5a, and four of the five yDm8 neurons (E, F, J and K) make more synapses onto Tm5a than Tm5b (yDm8-A, however, makes 10 synapses onto Tm5b-home and only five onto Tm5a-A; *Figure 3*). Each Dm9 receives input from both types of R7 and Dm8, and makes output synapses onto multiple R7 neurons (*Figure 3—source data 1*).

R7 synapses are polyadic, and Dm8 and Tm5a/b often sit together at synaptic sites. *Figure 4D and E* show R7 T-bars (active zone elements marking output synapses) adjacent to both yDm8 and Tm5a, and pDm8 and Tm5b, respectively. In column E, the yR7-E axon terminal, yDm8-E sprig, and Tm5a-E dendritic branch are tightly wrapped around each other (*Figure 4A,B*). yR7-E T-bars are distributed in layers M4-M6, and are apposed to postsynapses in the yDm8-E sprig and the distal dendritic branch of the Tm5a-E neuron (*Figure 4B*). Most yDm8-E output synapses are onto Tm5a-E and Dm9 (*Figure 3* and *Figure 3—source data 1*), and these are distributed between the sprig in M4-M5 and the main arbor in M6 (*Figure 4B* and associated *Figure 4—videos 1* and *2*). The only pDm8 with one home column in the reconstructed volume is pDm8-D. Although pDm8 usually have robust

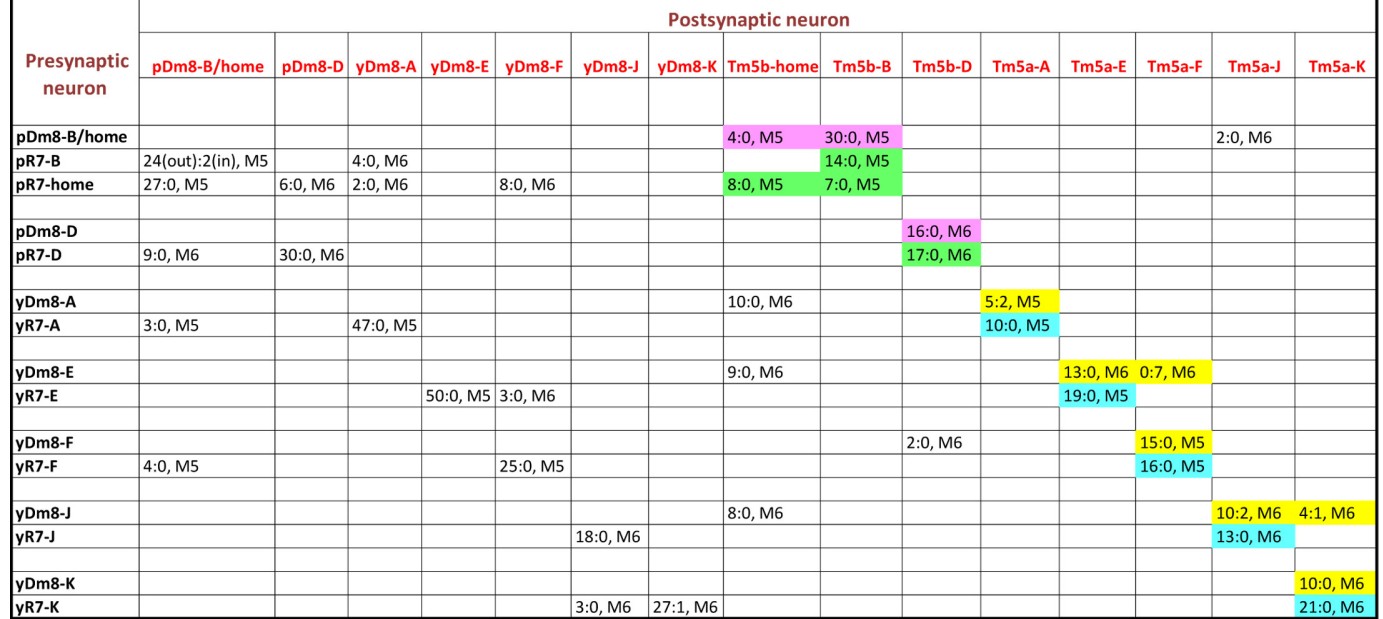

## Synaptic connections among R7, Dm8 and Tm5a/b

| Presynaptic neuron | pDm8-B/home | pDm8-D | yDm8-A | yDm8-E | yDm8-F | yDm8-J | yDm8-K | Tm5b-home | Tm5b-B | Tm5b-D | Tm5a-A | Tm5a-E | Tm5a-F | Tm5a-J | Tm5a-K |
|---|---|---|---|---|---|---|---|---|---|---|---|---|---|---|---|
| **pDm8-B/home** | | | | | | | | 4:0, M5 | 30:0, M5 | | | | | 2:0, M6 | |
| **pR7-B** | 24(out):2(in), M5 | | 4:0, M6 | | | | | | 14:0, M5 | | | | | | |
| **pR7-home** | 27:0, M5 | 6:0, M6 | 2:0, M6 | | 8:0, M6 | | | 8:0, M5 | 7:0, M5 | | | | | | |
| **pDm8-D** | | | | | | | | | | 16:0, M6 | | | | | |
| **pR7-D** | 9:0, M6 | 30:0, M6 | | | | | | | | 17:0, M6 | | | | | |
| **yDm8-A** | | | | | | | | 10:0, M6 | | | 5:2, M5 | | | | |
| **yR7-A** | 3:0, M5 | | 47:0, M5 | | | | | | | | 10:0, M5 | | | | |
| **yDm8-E** | | | | | | | | 9:0, M6 | | | | 13:0, M6 | 0:7, M6 | | |
| **yR7-E** | | | | 50:0, M5 | 3:0, M6 | | | | | | | 19:0, M5 | | | |
| **yDm8-F** | | | | | | | | | | | 2:0, M6 | | 15:0, M5 | | |
| **yR7-F** | 4:0, M5 | | | | 25:0, M5 | | | | | | | | 16:0, M5 | | |
| **yDm8-J** | | | | | | | | 8:0, M6 | | | | | | 10:2, M6 | 4:1, M6 |
| **yR7-J** | | | | | | 18:0, M6 | | | | | | | | 13:0, M6 | |
| **yDm8-K** | | | | | | | | | | | | | | | 10:0, M6 |
| **yR7-K** | | | | | | 3:0, M6 | 27:1, M6 | | | | | | | | 21:0, M6 |

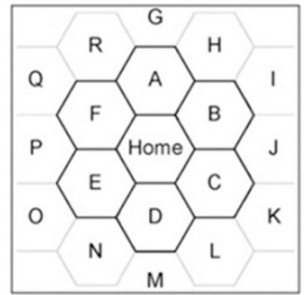
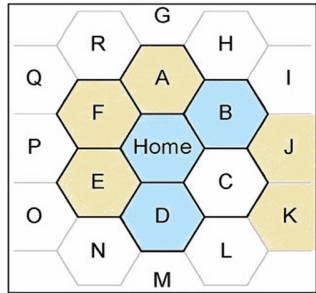

Map of arrangement of columns in EM reconstruction

**Figure 3.** Synaptic connections among R7, Dm8, and Tm5a/b from the EM reconstruction. The entries in column one indicate the EM column and y/p identities for each R7 and Dm8. The entries in row one indicate the column and y/p identities for the Dm8, Tm5a, and Tm5b neurons that are postsynaptic (and sometimes presynaptic) to the R7 and Dm8 neurons. In each box, the x:y nomenclature indicates the numbers of output synapses and input synapses. The layer in which most of these synapses are located is also indicated (M5 or M6). Classes of synapses are indicated in different colors, as follows: pR7→Tm5b, green; pDm8→Tm5b, purple; yR7→Tm5a, aqua; yDm8→Tm5a, yellow. At the bottom are maps of the arrangement of columns in the reconstruction, from *Takemura et al. (2015)*, with y columns indicated in gold and p columns in blue in the right-hand map. We could not assign y/p identities to columns C, P, or Q, and there are no R7 outputs listed for the other columns. Tm5-C has an ambiguous morphology and could be either a Tm5a or a Tm5b, and R7-P and R7-Q have no listed synapses onto Tm5a or b. *Figure 3—source data 1* contains additional information for non-home-column Dm8 inputs, and synapses onto Tm5c and Dm9.

The online version of this article includes the following source data for figure 3:

**Source data 1.** This is a table listing non-home column R7 synapses, and R7/Dm8 synapses with Tm5c and Dm9, in addition to the information in *Figure 3*.

sprigs (e.g. *Figure 2C*), pDm8-D has a very thin sprig, and most pR7-D and pDm8-D T-bars are in M6 (*Figure 4A,C* and associated *Figure 4—videos 3* and *4*).

Columns B and home form a two-home column pale circuit (*Figure 3*). The cells in this circuit are pR7-B, pR7-home, pDm8-B/home (a two-home column pDm8), Tm5b-B, and Tm5b-home (*Figure 4—figure supplement 1*). The B sprig of pDm8-B/home and one of the dendritic branches of Tm5b-B are both wrapped around the pR7-B terminal, while the pR7-home

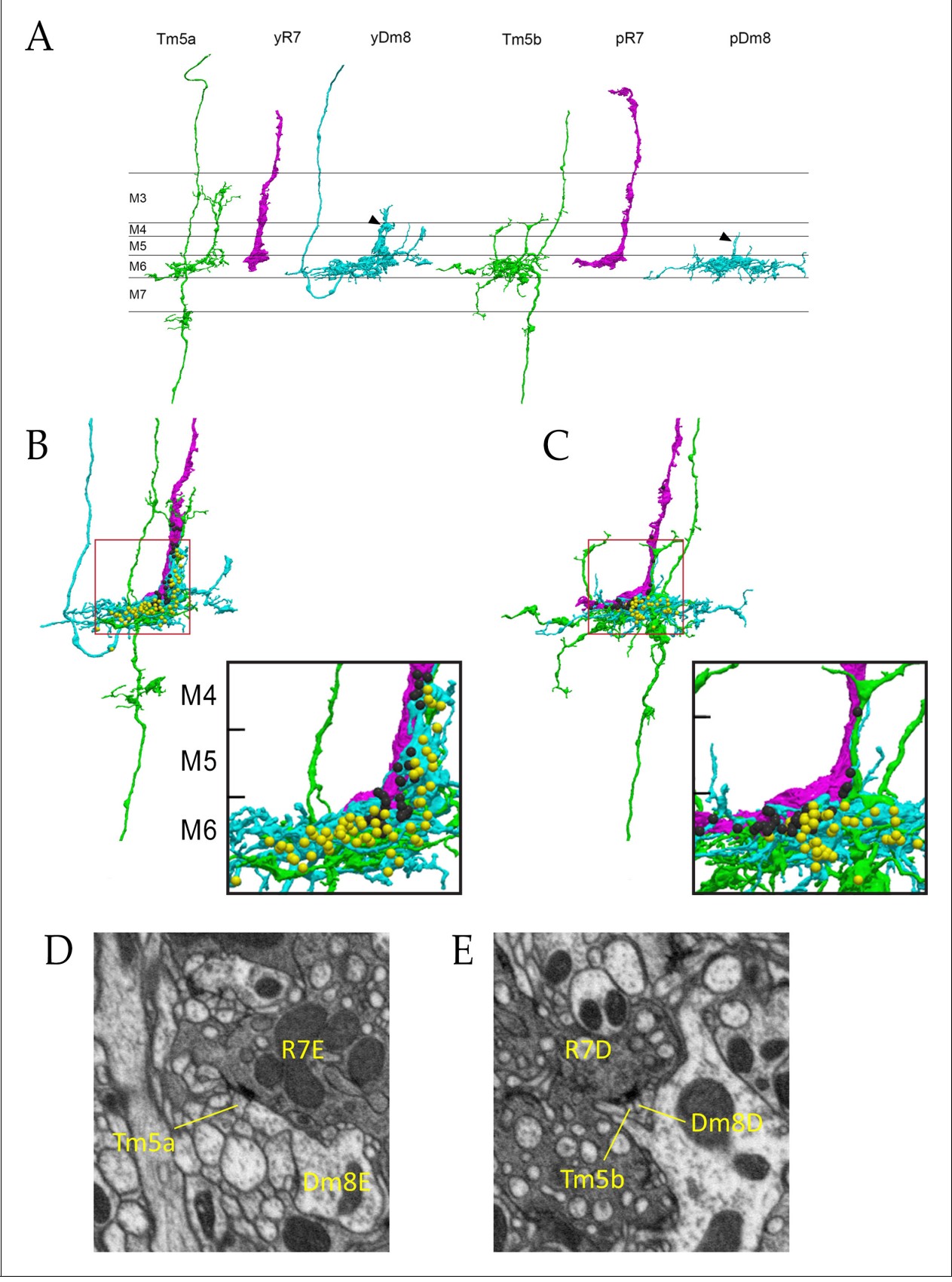

**Figure 4.** Electron microscopic reconstruction of wavelength discrimination circuits. Renderings of separated and combined cells from the EM reconstruction (*Takemura et al., 2015*) are shown in (A-D). R7, magenta; Dm8, cyan; Tm5a/b, green. (A) Separated cells in a yellow circuit, from column E, and a pale circuit, from column D. yR7-E, yDm8-E, and Tm5a-E pR7-D, pDm8-D, and Tm5b-D are shown. yDm8-E and pDm8-D sprigs indicated by arrowheads. (B-C) Renderings of R7-Dm8-Tm5a/b circuits. The insets show the home column region where most synapses are located. Black balls, R7 T-bars; yellow balls, Dm8 T-bars. The borders of M6, M5, and M4 are indicated. (B) The column E circuit. (C) The column D circuit. See also associated *Figure 4—videos 5* and *6* (vertical and horizontal rotation of each of these circuits, as well as of the two-home column B/home circuit, which is shown in *Figure 4—figure supplement 1*). For a comparison of an ExM image of a wild-type yR7 and yDm8 to yR7-E and yDm8-E from the EM reconstruction, see *Figure 6—figure supplement 1*. (D) A section from column E, showing a polyadic synapse of yR7-E onto yDm8-E and Tm5a-E. (E) A section from column D, showing a polyadic synapse of pR7-D onto pDm8-D and Tm5b-D. In (D) and (E), the R7 T-bars are the black shapes on the R7 membranes where they are apposed to both postsynaptic cells.

The online version of this article includes the following video and figure supplement(s) for figure 4:

**Figure supplement 1.** The columns B and home circuit.
**Figure 4—video 1.** Horizontal rotation of column E yellow circuit Colors as in *Figure 4*.
https://elifesciences.org/articles/48935#fig4video1
**Figure 4—video 2.** Vertical rotation of column E yellow circuit.
https://elifesciences.org/articles/48935#fig4video2
**Figure 4—video 3.** Horizontal rotation of column D pale circuit.
https://elifesciences.org/articles/48935#fig4video3
**Figure 4—video 4.** Vertical rotation of column D pale circuit.
https://elifesciences.org/articles/48935#fig4video4
**Figure 4—video 5.** Horizontal rotation of column B/home pale circuit.
https://elifesciences.org/articles/48935#fig4video5
**Figure 4—video 6.** Vertical rotation of column B/home pale circuit.
https://elifesciences.org/articles/48935#fig4video6

---

terminal is more loosely associated with the home sprig of pDm8-B/home and dendritic branches of both Tm5b neurons (*Figure 4—figure supplement 1*, and *Figure 4*-associated *Figure 4—videos 5* and *6*). All pR7-B→Tm5b synapses (14) and most pDm8-B/home→Tm5b synapses (30 *vs.* 4) are onto Tm5b-B. pR7-home makes similar numbers of synapses onto Tm5b-B and Tm5b-home (*Figure 3*).

## *DIP-γ* and *dpr11* mutations cause abnormalities in yDm8 dendritic arbors

If the yellow circuit is constructed using DIP-γ-Dpr11 interactions, as suggested by our analysis of the EM reconstruction, one might expect that there would be abnormalities in yDm8 arbors in *DIP-γ* and *dpr11* mutants. To assess yDm8 arbor morphology in whole-animal mutants, we generated flipouts in wild-type and mutants and examined arbor morphology (*Figure 5A–F*; *Figure 5—source data 1*). The sprig, which is located in the center of the arbor, usually has an expanded region at the distal end in M4/M5 (*Figure 5A–B*; see also *Figure 2A–C*). There is a large variation in sprig diameter and height in both wild-type and mutants. Maximum sprig diameter was significantly decreased in both *DIP-γ* and *dpr11* mutants compared to control, but the height of the sprig was relatively unaffected (*Figure 5C–F,I,K*). The numbers of two-home column single cell clones (with two sprigs) were higher in *DIP-γ* mutants as compared to controls (*Figure 5G–H*). We also examined contacts made by the yDm8 arbor to neighboring columns (*Figure 5J,L*). There was no difference in the percentage of yR7 columns contacted in either mutant, but yDm8 neurons in the *DIP-γ* mutant displayed an increase over control in the total number of columns contacted.

## yDm8 arbor morphology in detail

To obtain high-resolution views of interactions between yDm8 sprigs and R7 terminals, we examined single cell yDm8 clones in wild-type and *DIP-γ* mutant brains using expansion microscopy (ExM) (*Figure 6* and associated *Figure 6—videos 1–4*) (*Mosca et al., 2017*; reviewed by *Karagiannis and Boyden, 2018*). The morphology of the arbor in M6 and of the sprig in M4 and M5 can be visualized in detail in ExM images (*Figure 6A*). The sprig is on a home column R7, where most of the synapses from yR7 to yDm8 are located (*Figure 6B–C*, *Figure 6—figure supplement 1*).

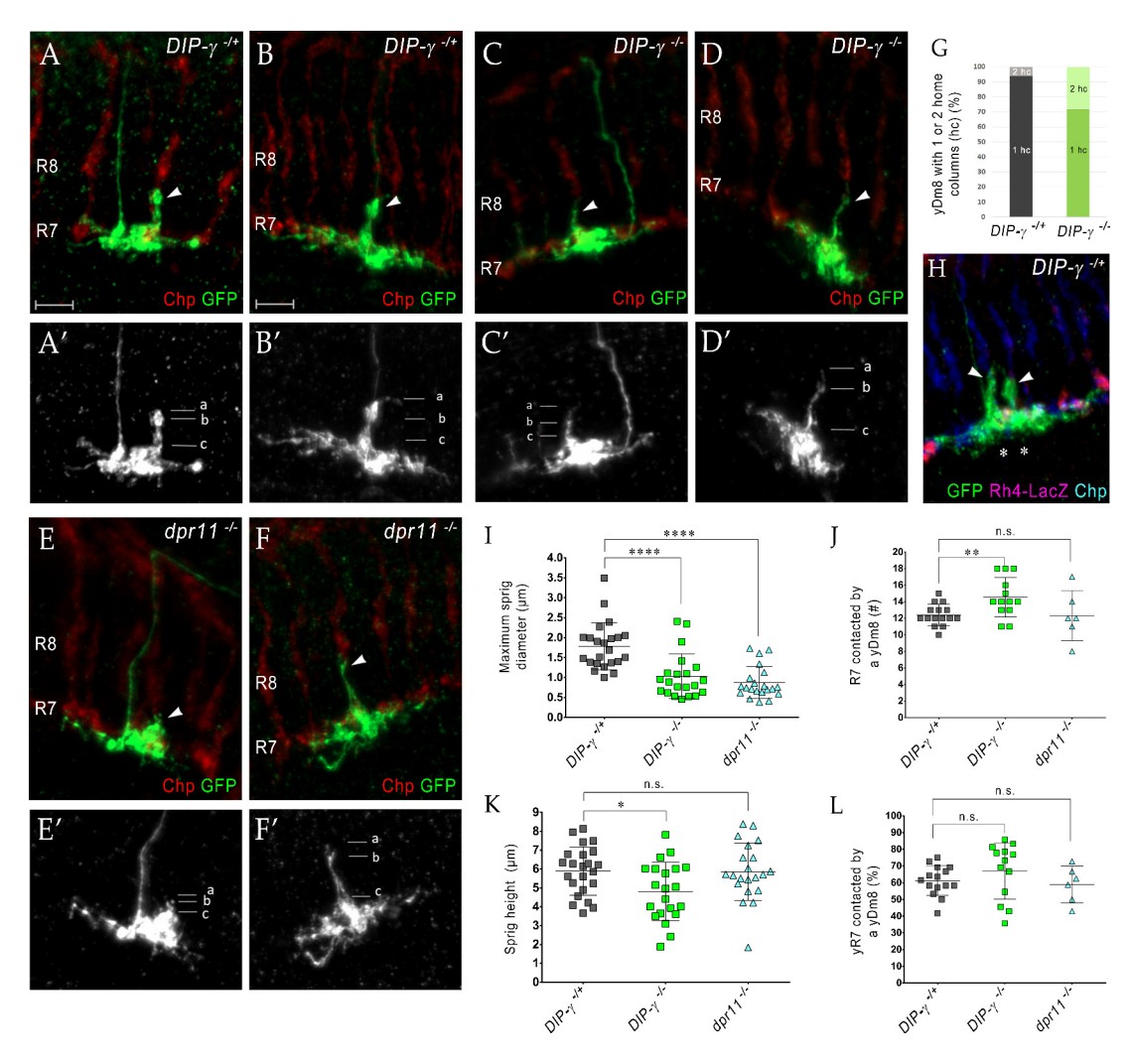

**Figure 5.** Loss of DIP-γ or Dpr11 alters morphology of yDm8 distal dendrites. (A–F) yDm8 flipouts generated in (A–B) wild-type, (C–D) *DIP-γ* mutant and (E–F) *dpr11* mutant. Sprigs are marked with arrowheads and PRs labeled with anti-Chp (red). (A'–F') Positions of the measurements used in graphs I and K are indicated. Height of the sprig was determined by measuring distance from 'a' to 'c' and sprig diameter was measured at position marked 'b'. Frequency of clones obtained for all three genotypes in *Figure 5—source data 1*. Flipouts in wild-type and *DIP-γ* mutants were generated with the yDm8 split-Gal4 driver; *dpr11* mutant flipouts were generated with the pan-Dm8 driver and scored as yDm8 using Rh4-LacZ labeling. Complete genotypes in *Table 2* of Materials and methods (*Table 1*). Maximum intensity projection; scale bar 5 μm. (G–H) Two-home column yDm8 flipouts were observed more frequently (4.5-fold increase) in *DIP-γ* mutant (11/39 clones) than in control (4/64 clones). Panel H shows a two home column yDm8 with both sprigs (arrowheads) located on yR7 columns (asterisks). (I, K) Maximum sprig diameter is reduced significantly in both *dpr11* and *DIP-γ* mutants. Sprig height was slightly affected in *DIP-γ* mutants but not in *dpr11* mutants. Graph shows mean +/- std. deviation and unpaired Student's t-test p-values. Sprig diameter: *DIP-γ* $^{-/+}$ 1.8+/- 0.6 (n = 23), *DIP-γ* $^{-/-}$1.0+/- 0.6 (n = 21), *dpr11* $^{-/-}$ 0.9+/- 0.4 (n = 21), ****p<0.0001 for both mutants. Sprig height: *DIP-γ* $^{-/+}$ 5.9+/- 1.3 (n = 23), *DIP-γ* $^{-/-}$4.8+/- 1.5 (n = 21), *dpr11* $^{-/-}$ 5.9+/- 1.5 (n = 21), from left to right *p=0.014, not significant (n.s.) (J, L) Contact with neighboring columns is affected in *DIP-γ* mutants. (J) Total number of R7 columns contacted by a yDm8: *DIP-γ* $^{-/+}$ 12.4+/- 1.3 (n = 15), *DIP-γ* $^{-/-}$14.5+/- 2.4 (n = 13), *dpr11* $^{-/-}$ 12.33+/- 3.0 (n = 6), from left to right **p=0.0061, not significant (n.s.) (L) Percentage of yR7 columns contacted by a yDm8: *DIP-γ* $^{-/+}$ 61.3+/- 8.9 (n = 15), *DIP-γ* $^{-/-}$67.03+/- 16.8 (n = 13), *dpr11* $^{-/-}$ 58.9+/- 11.0 (n = 6), from left to right, not significant (n.s.). The online version of this article includes the following source data for figure 5:

**Source data 1.** Frequency of two-home column yDm8 clones is increased in *DIP-γ* mutants.

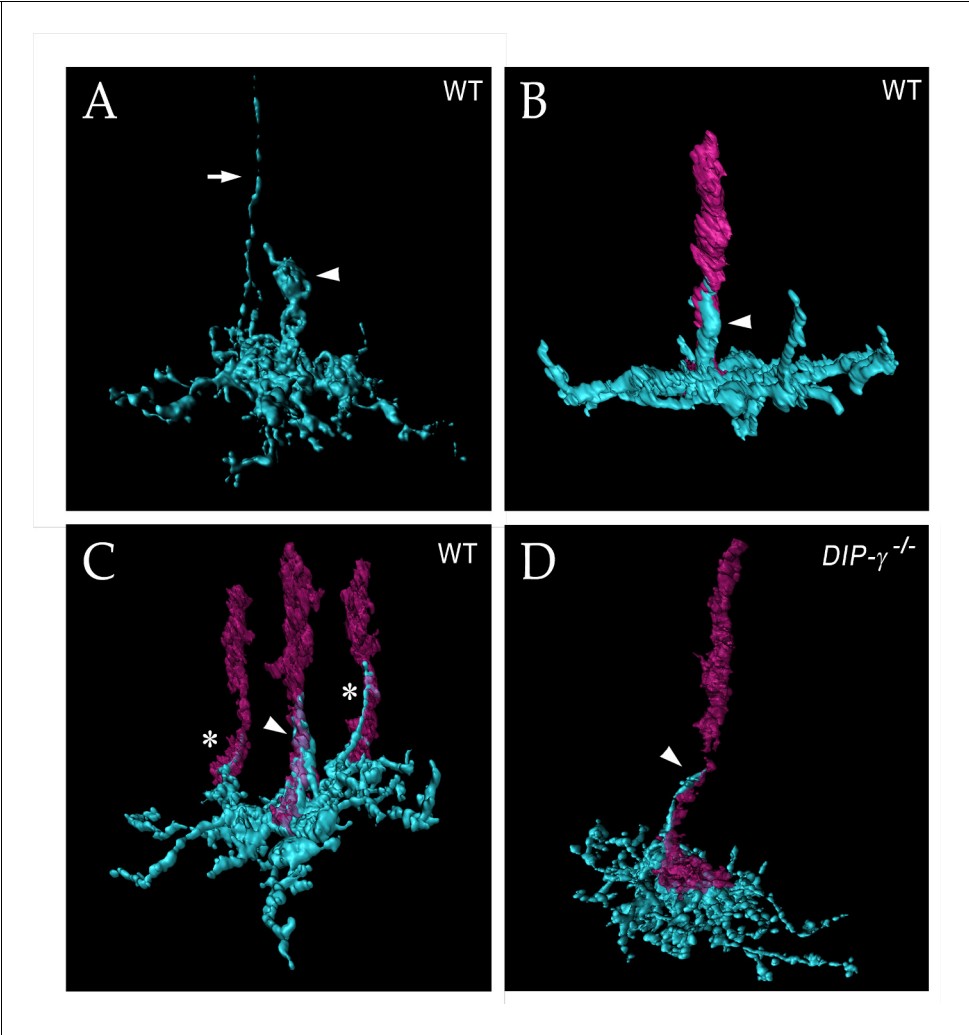

**Figure 6.** yDm8 arbor morphology in wild-type and *DIP-γ* mutant visualized with expansion microscopy. yDm8 dendritic arbors (cyan) visualized with expansion microscopy and surface rendered with Imaris software. (**A-C**) Wild-type; (**D**) DIP-γ $^{-/-}$. Arrowheads, sprigs; arrow in (**A**), axon. In (**B-D**), one or more R7 terminals/axons are included in the rendering. Two different views of the same flipout clone are shown in panels (**B**) and (**C**). The R7 terminals/axons are semi-transparent in (**C**) and (**D**). The home column R7 located at the center of the arbor makes extensive contacts with the sprig as well as with the base of the dendritic arbor in M6 in (**B**) and (**C**). Two thinner dendritic processes positioned on the edges of the arbor (asterisks) contact two non-home column R7 in (**C**). The yDm8 in the *DIP-γ* $^{-/-}$ mutant has a much thinner sprig as compared to wild-type (**D**). See associated ***Figure 6—videos 1–4*** (vertical and horizontal rotations) and ***Figure 6—figure supplement 2*** for additional views of the wild-type clone in (**C**) and the mutant clone in (**D**). ExM analysis of yDm8 in wild-type (n = 6) and DIP-γ $^{-/-}$ (n = 5) genotypes.

The online version of this article includes the following video and figure supplement(s) for figure 6:

**Figure supplement 1.** A comparison of the (**A**) ExM rendering of the wild-type yDm8 and yR7 shown in ***Figure 6B***, and the (**B**) EM reconstruction of yDm8-E and yR7-E shown in ***Figure 2A***.

**Figure supplement 2.** Additional views of expanded yDm8 in (**A**) wild-type and (**B**) *DIP-γ* $^{-/-}$.

**Figure 6—video 1.** Horizontal rotation of an expanded yDm8 in wild-type.
https://elifesciences.org/articles/48935#fig6video1

**Figure 6—video 2.** Vertical rotation of an expanded yDm8 in wild-type.
https://elifesciences.org/articles/48935#fig6video2

**Figure 6—video 3.** Horizontal rotation of an expanded yDm8 in *DIP-γ* mutant.
https://elifesciences.org/articles/48935#fig6video3

**Figure 6—video 4.** Vertical rotation of an expanded yDm8 in *DIP-γ* mutant.
https://elifesciences.org/articles/48935#fig6video4

In addition to the sprig, this yDm8 arbor has two thin dendritic processes that emerge from the base and are wrapped around neighboring non-home column R7 terminals (*Figure 6C*). The home column R7 extends into the base of the sprig, making multiple contacts with the base of the Dm8 arbor. *DIP-γ* mutant yDm8 neurons have thinner sprigs, in agreement with our quantitative analysis of mutant flipout clones from unexpanded samples (*Figures 5I* and *6D*; *Figure 6—figure supplement 2B*).

## Interactions between Dpr11 in yR7 and DIP-γ in yDm8 are required for yDm8 survival

We previously showed that Dm8 cells are lost in *DIP-γ*$^{Mi>GFP}$/*Df* animals (*Carrillo et al., 2015*). To explore this in detail, we used CRISPR-generated null mutants for both *DIP-γ* and *dpr11* (*Xu et al., 2018*) and determined yDm8 and pDm8 populations (*Figure 7A–C*). *DIP-γ*$^{Mi>GFP}$, which has no detectable protein expression, was used as one of the *DIP-γ* alleles to allow identification of yDm8 soma. Both *DIP-γ* and *dpr11* mutants showed a ~ 50% decrease in yDm8 cell number using the pan-Dm8 driver and 60–65% loss when determined with Dac (*Figure 7A,C*). A double mutant had a similar phenotype to the two single mutants (*Figure 7A*), suggesting that the two genes function in the same pathway(s). The loss of yDm8 in *DIP-γ* mutants was partially rescued by expressing a *DIP-γ* transgene using the *DIP-γ Gal4* driver (*Figure 7A*). To determine if yDm8 loss in *dpr11* mutants was due to the absence of Dpr11 from the eye or from other Dpr11-expressing cells in the medulla, we performed eye-specific *dpr11* transgenic RNAi, and found a ~ 40% reduction in yDm8 cell numbers (*Figure 7C*).

We also ectopically expressed DIP-γ in PRs, and found that this phenocopied the *DIP-γ* and *dpr11* loss-of-function (LOF) phenotypes (*Figure 7C*). This result suggests that the presence of both Dpr11 and its partner on the same cells (expression in cis) prevents Dpr11 in yR7 from interacting with DIP-γ in trans (on yDm8). Ectopic expression of DIP-α, a different DIP that does not bind to Dpr11, had no effect on yDm8 survival (*Figure 7C*), indicating that the Dpr11-DIP-γ interaction is required for yDm8 selection and survival. Similar results were observed for DIP-α and Dpr10 in the neuromuscular system, where DIP-α is normally expressed on the MNISN-1s (RP2) motoneuron and Dpr10 on its muscle targets. LOF mutants lacking either protein are missing specific muscle branches, and *cis* ectopic expression of DIP-α on muscles, or of Dpr10 on MNISN-1s, generates the same phenotypes (*Ashley et al., 2019*).

To assess whether the decrease in yDm8 population was due to apoptotic cell death, we expressed *Drosophila* inhibitor of apoptosis protein 1 (DIAP) in *DIP-γ* expressing cells (*Figure 7—figure supplement 1*). DIAP overexpression rescued the loss of yDm8 in *DIP-γ* mutants, indicating that DIP-γ is required for cell survival and that yDm8 undergo apoptosis when it is absent (*Figure 7A*). Furthermore, the number of yDm8 that survived when DIAP was expressed was ~45% greater than the number of yDm8 present in controls. It has been reported that extensive cell death occurs in the medullary cortex in wild-type during normal optic lobe development (*Togane et al., 2012*). Thus, in addition to the yDm8 that were rescued from cell death caused by the absence of DIP-γ, DIAP also rescued yDm8 that were lost due to normal developmental cell death (see Discussion). The fact that *dpr11* and *DIP-γ* mutants displayed a similar extent of yDm8 loss implies that Dpr11 in yR7 signals to yDm8 *via* DIP-γ to ensure their survival. Absence of either molecule compromises the interaction and leads to death of yDm8 neurons.

We next assessed pDm8 populations in the above genotypes, including double mutants of *dpr11* and *DIP-γ*, and found that they did not differ significantly from controls. Importantly, pDm8 cell numbers did not decrease when the yDm8 population was increased by DIAP rescue, indicating that the two populations of Dm8 neurons do not normally compete for survival signals (*Figure 7B*; see below).

We also visualized the M6 layer using the *DIP-γ*$^{Mi>GFP}$ reporter and found that the morphologies of the yDm8 layer in the above genotypes were consistent with the results obtained by counting soma in the cortex. In the neuropil, larger gaps were observed in the mutants, and fewer and smaller gaps in wild-type and rescue genotypes (*Figure 7D–H*).

*DIP-γ* mutants also displayed a ~ 4.5-fold higher frequency of two-home column flipouts as compared to control (*Figure 5G*). The increase in two-home column yDm8 could be a response strategy in which yDm8 neurons innervate more yR7 PRs in order to compensate for the loss of survival signals mediated by DIP-γ.

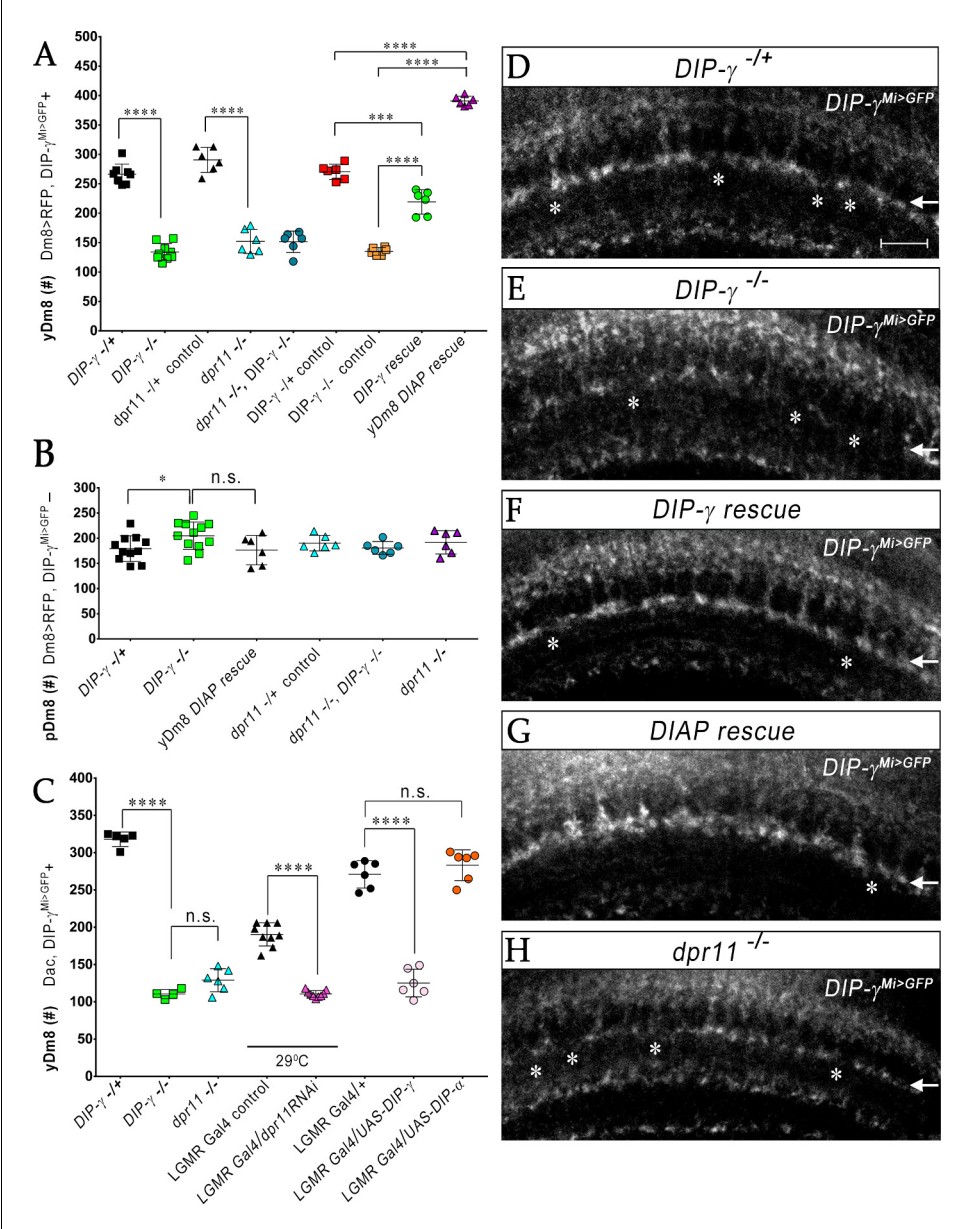

**Figure 7.** yR7 PRs provide survival signals to yDm8 neurons through Dpr11-DIP-γ interactions. (A–B) yDm8 and pDm8 cell numbers were determined using pan-Dm8 driver >RFP and *DIP-γ^{Mi>GFP}* reporter. yDm8 neurons express both RFP and GFP, and pDm8 neurons express only RFP. (C) yDm8 cell number determined using anti-GFP for *DIP-γ^{Mi>GFP}* reporter and anti-Dac. The *DIP-γ^{Mi>GFP}* reporter (indicated on the x-axis in all graphs as *DIP-γ^{-/+}*) is an insertion in the 5' UTR intron that has no detectable protein expression. Thus, this line serves as a mutant as well as a reporter of *DIP-γ* transcript expression. Graphs show mean +/- std. deviation and unpaired Student's t-test p-values. Complete genotypes in *Table 2* of Materials and methods (*Table 1*). (A) Both *DIP-γ* and *dpr11* mutants show ~50% loss of yDm8 neurons, and this is rescued in *DIP-γ* mutants by inhibiting cell death with DIAP. yDm8 cell numbers in heterozygous controls, *DIP-γ* and *dpr11* mutants, a double mutant of both genes, and *DIP-γ* and DIAP rescues in *DIP-γ* mutant are shown. *DIP-γ^{-/+}* 266.4+/- 17.2 (n = 8), *DIP-γ^{-/-}* 134+/- 14.3 (n = 9), ****p<0.0001; *dpr11^{-/+}* control 290.7+/- 21.3 (n = 6), *dpr11^{-/-}* 152+/- 20.4 (n = 6), ****p<0.0001; *dpr11^{-/-}*, *DIP-γ^{-/-}* 151+/- 18.3 (n = 6), *DIP-γ^{-/-}* 134+/- 14.3 (n = 6), not significant; *dpr11^{-/-}*, *DIP-γ^{-/-}* 151+/- 18.3 (n = 6), *dpr11^{-/-}* 152+/- 20.4 (n = 6), not significant; *DIP-γ^{-/+}* control 270+/- 13.03 (n = 6), *DIP-γ* rescue 219.2+/- 20.7 (n = 6), ***p=0.0004; *DIP-γ^{-/-}* mutant control 135.2+/- 6.5 (n = 6), *DIP-γ* rescue 219.2+/- 20.7 (n = 6), ****p<0.0001; *DIP-γ^{-/+}* control 270+/- 13.03 (n = 6), yDm8 *DIAP* rescue 390.8+/- 8.0 (n = 6), ****p<0.0001; *DIP-γ^{-/-}* mutant control 135.2+/- 6.5 (n = 6), yDm8 *DIAP* rescue 390.8+/- 8.0 (n = 6), ****p<0.0001. (B) pDm8 numbers are unchanged in mutants and in DIAP rescue. *DIP-γ^{-/+}* 179.4+/- 25.6 (n = 11), *DIP-γ^{-/-}* 204.9+/- 27.2 (n = 12), *p=0.03; *dpr11^{-/+}* control 176.5+/- 29.2 (n = 6), *dpr11^{-/-}* 190.2+/- 15.5 (n = 6), not significant (n.s.); *dpr11^{-/-}*, *DIP-γ^{-/-}* 180.7+/- 12.8 (n = 6), *DIP-γ^{-/-}* 204.9+/- 27.2 (n = 12), not significant; *dpr11^{-/-}*, *DIP-γ^{-/-}* 180.7+/- 12.8 (n = 6), *dpr11^{-/-}* 190.2+/- 15.5, not significant; *DIP-γ^{-/-}* 204.9+/- 27.2 (n = 12), yDm8 *DIAP* rescue 191.8+/- 23.2 (n = 6), not significant (n.s.). (C) Dpr11 in R7 is required for yDm8 survival. yDm8 cell number in wild-type, mutants, *dpr11* eye-specific RNAi, and ectopic DIP-γ expression in PRs. *DIP-γ^{-/+}* 318+/- 9.7 (n = 5), *DIP-γ^{-/-}* 110.5+/- 6.1 (n = 4), ****p<0.0001; *DIP-γ^{-/-}* 110.5+/- 6.1 (n = 4), *dpr11^{-/-}* 129+/- 15.4 (n = 6), not significant (n.

*Figure 7 continued on next page*

Figure 7 continued

s.); *lGMR-Gal4* control at 29°C 190.3+ /- 15.5 (n = 9), *lGMR-Gal4 >dpr11 RNAi* 110.6+ /- 4.5 (n = 9), ****p<0.0001; *lGMR-Gal4* control 271+ /- 18.3 (n = 6), *lGMR-Gal4 >UAS-DIP-γ* (n = 6) 125.2+ /- 18.5, ****p<0.0001; *lGMR-Gal4* control 271+ /- 18.3 (n = 6), *lGMR-Gal4 >UAS-DIP-α* 283.2+ /- 20.6 (n = 6), not significant (n.s.). (D-H) yDm8 labeling in the neuropil in wild-type, *DIP-γ* mutant, *DIP-γ* rescue, *DIAP* yDm8 rescue, and *dpr11* mutant, using *DIP-γ^{Mi>GFP}* reporter. Large gaps (asterisks) representing yDm8 cell death are seen in the M6 layer (arrow) in both *DIP-γ* and *dpr11* null mutants (E, H), whereas wild-type, *DIP-γ* and *DIAP* rescues showed smaller and fewer gaps (asterisks in D, F-G). Adult optic lobes were labeled with anti-GFP for yDm8 reporter.

The online version of this article includes the following figure supplement(s) for figure 7:

**Figure supplement 1.** DIAP localizes to DIP-γ expressing cells.

## DIP-γ controls yDm8 death by interacting with Dpr11 during early pupal development

Having found that interactions between yR7 and yDm8 mediated by Dpr11 and DIP-γ are required for yDm8 survival in adults (*Figure 7*), we next determined when the interaction that ensures survival occurs. To this end, we examined yDm8 loss at different stages of OL development in wild-type and *DIP-γ* mutant animals starting at 15 hr. (15 hr) after puparium formation (APF) (*Figure 8A*). yDm8 populations in wild-type and mutant animals were examined by labeling with Dac and *DIP-γ^{Mi>GFP}* reporter at 15 hr, 25 hr and 45 hr APF. At 15 hr APF, yDm8 cell numbers in *DIP-γ* mutants are similar to those in wild-type, suggesting that the Dpr11-DIP-γ interaction is needed for yDm8 survival only after that time. In wild-type, yDm8 numbers do not change between 15 hr APF and 25 hr APF. Between 25 hr APF and 45 hr APF, about 30% of yDm8 soma are lost, and yDm8 numbers then remain stable until adulthood (*Figure 8A*). However, in *DIP-γ* mutants, more than 1/3 of the yDm8 soma are lost between 15 hr APF and 25 hr APF, and by 45 hr APF only 25% of the original number of yDm8 soma present at 15 hr APF remain (*Figure 8A*). As in wild-type, these numbers in *DIP-γ* mutants then remain stable until adulthood. Inhibition of cell death in the *DIP-γ* mutant with DIAP restored the population in the adult back to the original wild-type pool size present at 15 hr APF (*Figure 8A*). Thus, DIP-γ is required for suppression of yDm8 cell death early in pupal development. Extensive migration of yDm8 soma was observed in wild-type, and this migration was unaffected in the *DIP-γ* mutant (*Figure 8—figure supplement 2A–D*).

To detect cells that are undergoing apoptosis in *DIP-γ* mutants, we used Apoliner, a fluorescent reporter of caspase activity, to mark yDm8 at 25 hr APF in wild-type and mutant genotypes (*Bardet et al., 2008*) (*Figure 8B–E*). The Apoliner reporter consists of a membrane tethered RFP joined to GFP with a nuclear localization signal and a caspase site between the RFP and GFP moieties. In cells undergoing apoptosis, activated caspases cleave Apoliner, releasing GFP to localize to the nucleus (*Figure 8B*). The Apoliner transgene was driven by a *DIP-γ* split Gal4 driver in wild-type and *DIP-γ* mutant and yDm8 cell bodies were identified by labeling with Dac. At 24–25 hr APF, there were twice as many dying yDm8 cells in *DIP-γ* mutants as compared to control, further confirming the role of Dpr11-DIP-γ interaction in yDm8 cell survival (*Figure 8C–E*).

We next determined when Dpr11-expressing yR7 and DIP-γ expressing yDm8 meet in the medulla neuropil. Since yDm8 cell death caused by the loss of DIP-γ begins to occur between 15 hr and 25 hr APF (*Figure 8A–E*) and because R7 terminals do not mature till after the mid-pupal stage (*Ting et al., 2005*), we investigated if the Dpr11-DIP-γ interaction required for yDm8 survival occurred at the growth cone stage of R7 development. Indeed, we found that *dpr11* was expressed in select R7 growth cones at 15 hr APF, suggesting that the yellow fate of R7 was determined by that time (*Figure 8F*). We next examined *DIP-γ* expression in early pupa and found DIP-γ labeling adjacent to select R7 at 20 hr APF, indicating that yDm8 arbors are apposed to yR7 terminals (*Figure 8G*). The gaps in DIP-γ labeling are presumed to be pDm8 (*Figure 8—figure supplement 1*). Taken together, our results are consistent with a model in which selection of yDm8 by yR7 occurs by 15 hr - 20 hr APF *via* Dpr11-DIP-γ binding and thereby ensures survival of yDm8.

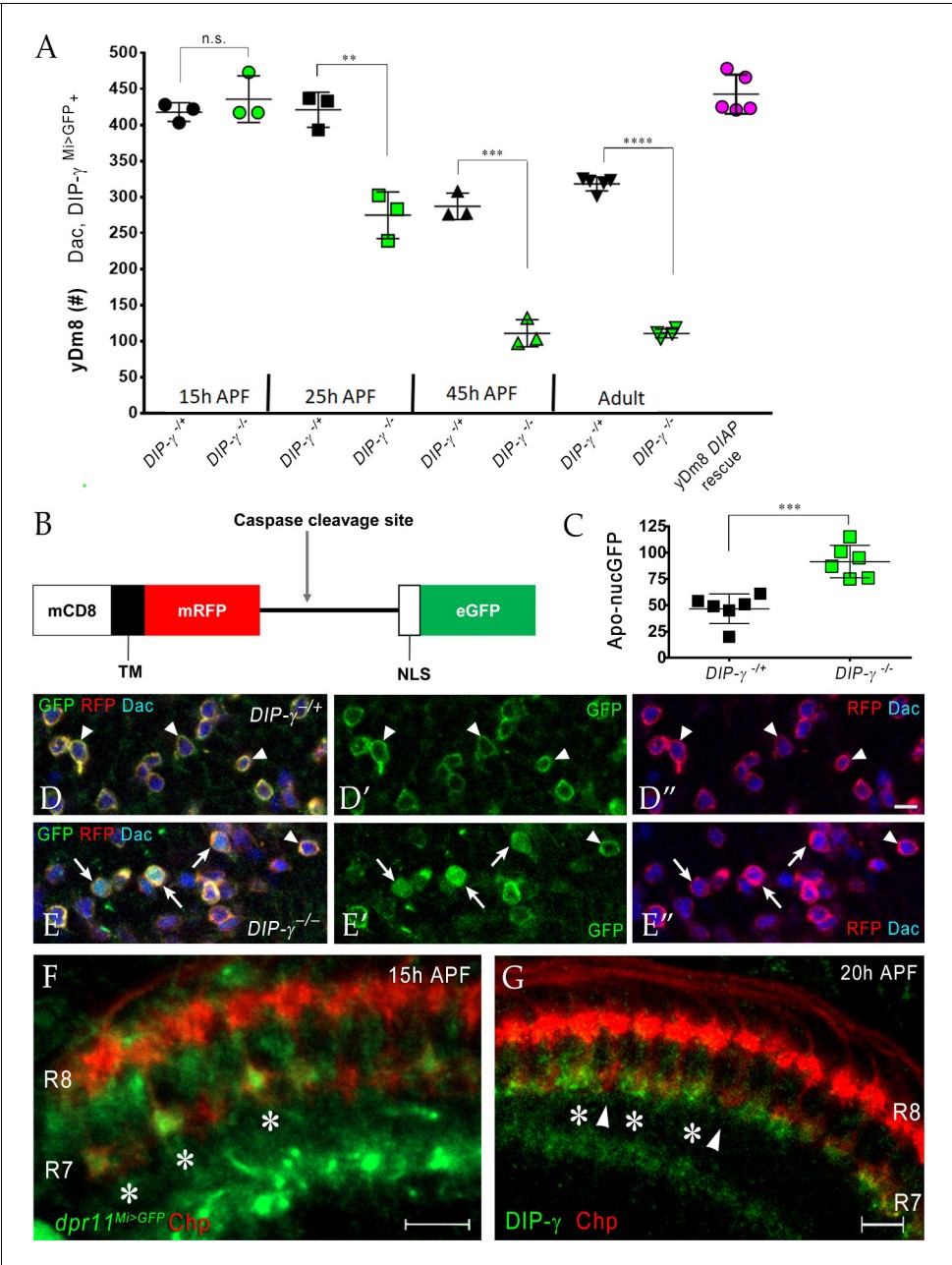

**Figure 8.** Dpr11-DIP-γ interactions are required early in pupal development to prevent yDm8 cell death. (A) yDm8 cell death occurs in *DIP-γ* mutants between 15 hr and 45 hr APF. yDm8 cell death in wild-type occurs between 25 hr and 45 hr APF (p-value below). DIAP expression in the *DIP-γ* mutant rescues cell number back to the original level at 15 hr APF (p-value below). yDm8 cell number determined with anti-GFP for *DIP-γ*$^{Mi>GFP}$ reporter and anti-Dac. *DIP-γ*$^{Mi>GFP}$ reporter heterozygote indicated as *DIP-γ*$^{-/+}$. Graph shows mean + /- std. deviation and unpaired Student's t-test p-values. Complete genotypes in *Table 2* of Materials and methods (*Table 1*). 15 hr APF: *DIP-γ*$^{-/+}$ 417.7+ /- 13.1 (n = 3), *DIP-γ*$^{-/-}$435.7+ /- 32.3 (n = 3), not significant (n.s.); 25 hr APF: *DIP-γ*$^{-/+}$ 421+ /- 24.3 (n = 3), *DIP-γ*$^{-/-}$274.7+ /- 32.3 (n = 3), **p=0.0033; 45 hr APF: *DIP-γ*$^{-/+}$ 287+ /- 18.2 (n = 3), *DIP-γ*$^{-/-}$110.7+ /- 18.7 (n = 3), ***p=0.0003; Adult: *DIP-γ*$^{-/+}$ 318+ /- 9.7 (n = 5), *DIP-γ*$^{-/-}$ 110.5+ /- 6.1 (n = 4), ****p<0.0001; p-values below are not shown on the graph: 25 hr APF: *DIP-γ*$^{-/+}$ 421+ /- 24.3 (n = 3), 45 hr APF: *DIP-γ*$^{-/+}$ 287+ /- 18.2 (n = 3), **p=0.0016; yDm8 *DIAP* rescue 442.6+ /- 27.2 (n = 5), 15 hr APF *DIP-γ*$^{-/+}$ 417.7+ /- 13.1, not significant (n.s.) 15 hr APF: *DIP-γ*$^{-/+}$ 417.7+ /- 13.1 (n = 3), Adult: *DIP-γ*$^{-/+}$ 318+ /- 9.7 (n = 5), ****p<0.0001; 15 hr APF: *DIP-γ*$^{-/-}$435.7+ /- 32.3 (n = 3), 25 hr *DIP-γ*$^{-/-}$274.7+ /- 32.3 (n = 3), **p=0.004; 25 hr APF: *DIP-γ*$^{-/-}$274.7+ /- 32.3 (n = 3), 45 hr *DIP-γ*$^{-/-}$110.7+ /- 18.7 (n = 3), **p=0.0016. (B-E) yDm8 cell bodies marked with apoptotic reporter Apoliner are significantly increased in *DIP-γ* mutant. (B) Schematic of Apoliner (*Bardet et al., 2008*). In live cells, GFP is tethered to the membrane with RFP. In dying cells, activated caspases cleave Apoliner to release the GFP moiety that localizes to the nucleus. (C) Quantitation of dying yDm8 cell bodies in control and *DIP-γ* mutant. Graph shows mean + /- std. deviation and unpaired Student's t-test p-values, n = 6 OLs for each genotype. Complete genotypes in *Table 2* of Materials and methods (*Table 1*). *DIP-γ*$^{-/+}$ 46.7+ /- 14.12 (n = 6), *DIP-γ*$^{-/}$ 91.5+ /- 15.4 (n = 6), ***p=0.0004. (D-E) Visualization of Apoliner in yDm8 soma in control (D) and *DIP-γ* mutant (E). UAS-
*Figure 8 continued on next page*

Figure 8 continued

Apoliner was driven with a *DIP-γ* split Gal4 driver and yDm8 soma were labeled with anti-Dac, anti-GFP and anti-RFP. Dying yDm8 with nuclear GFP are seen in the *DIP-γ* mutant (arrows in **E**), while live yDm8 with membrane-localized GFP and RFP are seen in both control and mutant (arrowheads). (**F**) Dpr11 is expressed in select R7 PRs at 15 hr APF (asterisks). *dpr11^{Mi>GFP}* reporter labeled with anti-Chp (red) and anti-GFP (green) at 15 hr APF. Note that one of the younger R7 PRs located on the right of the image also expresses Dpr11. Single confocal slice; scale bar 5 µm. (**G**) DIP-γ is expressed in yDm8 neurons apposed to specific R7 PRs by 20 hr APF (asterisks). pR7 terminals (arrowheads) do not show overlapping DIP-γ labeling (see also *Figure 8—figure supplement 1B–B'*). Wild-type, labeled at 20 hr APF with anti-DIP-γ (green) and anti-Chp (red). Complete genotypes in *Table 2* of Materials and methods. Single confocal slice; scale bar 10 µm.

The online version of this article includes the following figure supplement(s) for figure 8:

**Figure supplement 1.** Dpr11 and DIP-γ are expressed in yR7 and yDm8, respectively, around the time yR7 selects yDm8 for survival.

**Figure supplement 2.** yDm8 cell death in *DIP-γ* mutant during migration of the cell bodies.

## Perturbing R7 fate in the retina alters Dm8 fate in the medulla

Since Dpr11 is expressed in yR7 PRs, we next examined how the two subtypes of Dm8 neurons respond when R7 fates are altered. The transcription factor Spineless (Ss) controls yellow and pale subtype choice in the retina. Ectopic expression of *ss* in all PRs (*ss^{GOF}*) induces yellow fate in all R7 cells and prevents formation of pR7 (*Wernet et al., 2006*). We first determined Dpr11 and DIP-γ expression in *ss^{GOF}*. In contrast to wild-type, where only yR7 is labeled by *dpr11^{Mi>GFP}*, all R7 terminals showed GFP expression when *ss* was mis-expressed (*Figure 9A–B*). Thus, Ss expression activates Dpr11 along with Rh4 and other yR7-specific genes. The normal gaps in *DIP-γ^{Mi>GFP}* labeling in the M6 layer were missing in *ss^{GOF}* (compare 1A and C in *Figure 9—figure supplement 1*). When *DIP-γ* was removed in the *ss^{GOF}* background, gaps were again observed in the yDm8 layer, indicating that the M6 signal in *ss^{GOF}* is indeed due to DIP-γ-expressing yDm8 (*Figure 9—figure supplement 1A–C*). The change in R7 fate is thus transmitted to the downstream circuit in the medulla via Dpr11-DIP-γ interactions, resulting in uniform yDm8 labeling in the M6 layer.

We determined yDm8 and pDm8 cell number in the above genotypes (*Figure 9C–D*). In *ss^{GOF}*, we observed a 36% increase in yDm8, while pDm8 cell number was decreased by 84%. The increase in yDm8 in *ss^{GOF}* was dependent on DIP-γ, as absence of *DIP-γ* in the *ss^{GOF}* background resulted in 53% loss of yDm8 neurons. Interestingly, pDm8 increased significantly, doubling their cell numbers as compared to *ss^{GOF}* alone (*Figure 9D*). Thus, yDm8 and pDm8 populations compete with each other when R7 fate is altered.

To assess Dm8 subtypes in the reverse situation where there are only pR7 in the eye, we ectopically expressed the Defective proventriculus (Dve) transcription factor, a repressor downstream of *ss*, in all PRs (*Johnston et al., 2011*; *Nakagawa et al., 2011*; *Yan et al., 2017*). Expression of Dve in all PRs (*dve^{GOF}*) converts all ommatidia to pale (*Yan et al., 2017*). We used this strategy because *ss* LOF mutations cause lethality and we observed no phenotype with eye-specific *ss* RNAi. Since the external eye morphologies of *dve^{GOF}* animals were abnormal, we initially examined the neuropil layers and PR targeting in the medulla (*Figure 9—figure supplement 1E,K*). Except for the *DIP-γ^{Mi>GFP}* labeling in M6, other layers of the neuropil showed similar patterns to wild-type. The M6 layer had large gaps, similar to those observed in *DIP-γ* mutants, suggesting that there were fewer yDm8 neurons (*Figure 9—figure supplement 1E*). Indeed, yDm8 cell number decreased by ~60% as compared to the control (*Figure 9G*).

Sevenless is a receptor tyrosine kinase that is required for specification of R7 in the developing eye. In *sev* mutants, R7 PRs are absent, because the R7 precursor cell becomes a non-neuronal cone cell. We found that both the yDm8 and pDm8 populations were reduced in *sev* mutants by >70% as compared to control cell numbers, showing that both classes of Dm8 require interaction with R7 PRs for survival. yDm8 cell numbers were further reduced when DIP-γ was removed in the *sev* mutant background (*Figure 9C*). This dependence on DIP-γ likely occurs because there are a few R7 remaining in the *sev* mutant (*Figure 9—figure supplement 1F*), although the allele we used is described as an amorph. The ~50 yDm8 that remain in a *sev DIP-γ* double mutant may not require interactions with other cells for survival, or may obtain support through a different pathway that does not involve DIP-γ. pDm8 numbers are slightly increased by removal of DIP-γ in the *sev* mutant background. This is probably because they are competing with yDm8 for the few remaining R7 PRs, so that the loss of additional yDm8 through removal of DIP-γ frees up those slots to be occupied by pDm8.

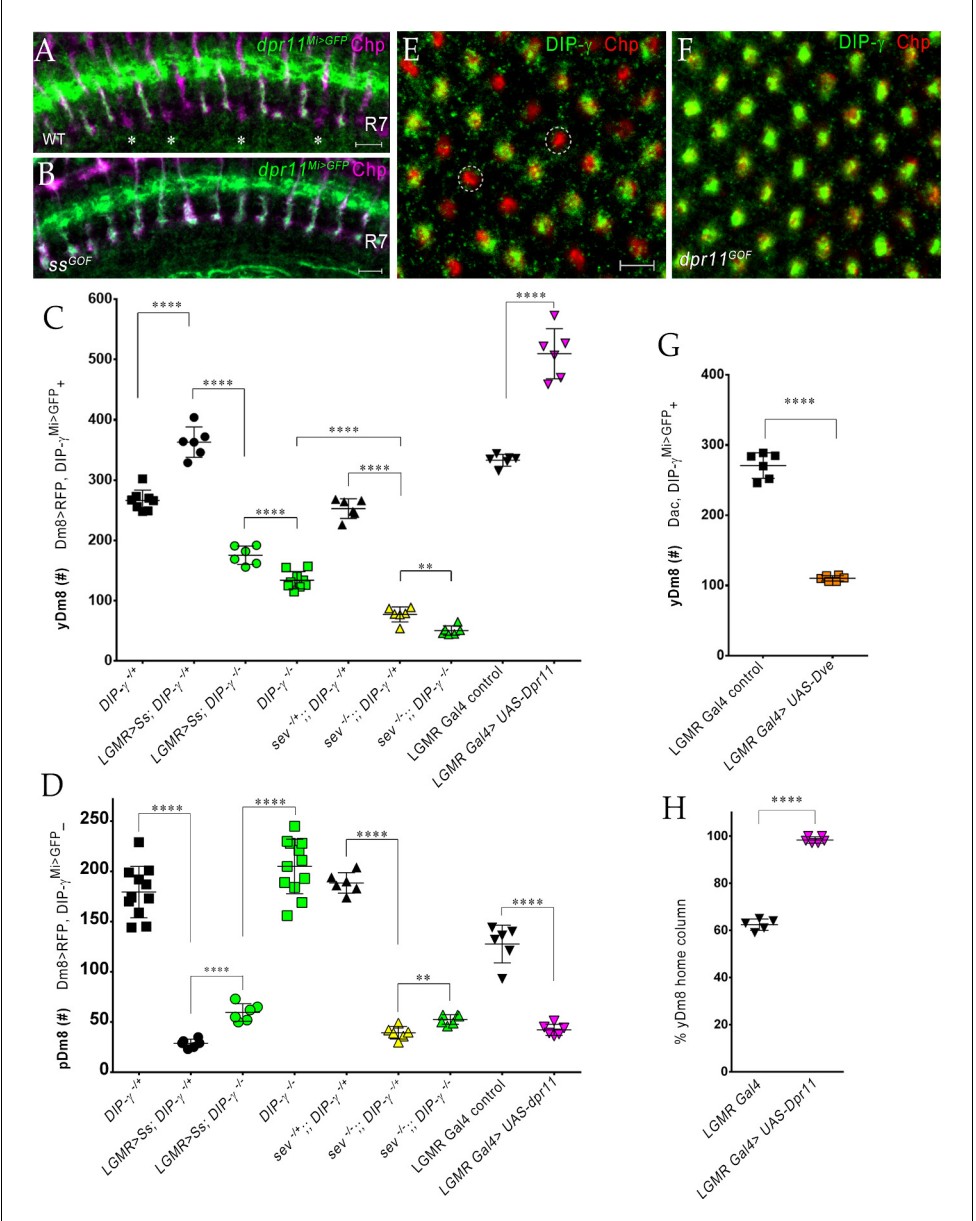

**Figure 9.** The representation of Dm8 subtypes in the medulla is altered by manipulation of R7 fates in the retina. (A–B) Conversion of all R7 PRs to the yR7 fate in *ss^GOF* results in Dpr11 expression in every R7. Dpr11 is expressed in yR7 PRs in (A) wild-type and (B) *ss^GOF*. pR7 PRs are present in wild-type (asterisk) and absent in *ss^GOF*. Mid-pupal optic lobes labeled with anti-GFP (green) for *dpr11^Mi>GFP* reporter and anti-Chp (magenta) for all PRs. Maximum intensity projection; scale bar 5 μm. (C–D) yDm8 and pDm8 cell number determined using pan-Dm8 driver >RFP and *DIP-γ^Mi>GFP* reporter. yDm8 expresses both RFP and GFP and pDm8 expresses only RFP. Graphs show mean + /- std. deviation and unpaired Student's t-test p-values. Complete genotypes in ***Table 2*** of Materials and methods (***Table 1***). *DIP-γ^Mi>GFP* reporter heterozygote indicated as *DIP-γ^-/+*. (C) More yDm8 neurons survive when all R7 PRs are converted to the yR7 fate (*ss^GOF*) or when they all express Dpr11 (*dpr11^GOF*). yDm8 neurons are also lost when R7 PRs are absent (*sev*) and/or *DIP-γ* is mutant. yDm8 cell number in wild-type, *ss^GOF*, *sev* mutant, *dpr11^GOF*: *DIP-γ^-/+* 266.4+ /- 17.2 (n = 8), *ss^GOF*, *DIP-γ^-/+* 363 + /- 25.3 (n = 6), ****p<0.0001; *ss^GOF*, *DIP-γ^-/+* 363 + /- 25.3 (n = 6), *ss^GOF*, *DIP-γ^-/-*175.3+ /- 15.2 (n = 6), ****p<0.0001; *DIP-γ^-/-*134+ /- 14.3 (n = 9), *ss^GOF*, *DIP-γ^-/-*175.3+ /- 15.2 (n = 6), ***p=0.0001; *sev^-/+*; ; *DIP-γ^-/+* 252.8+ /- 16.3 (n = 6), *sev^-/-*; ; *DIP-γ^-/+* 77.3+ /- 12.5 (n = 6), ****p<0.0001; *sev^-/-*; ; *DIP-γ^-/+* 77.3+ /- 12.5 (n = 6), *sev^-/-*; ; *DIP-γ^-/-* 50.5+ /- 7.8 (n = 6), **p=0.0012; *DIP-γ^-/-* 134+ /- 14.3 (n = 9), *sev^-/-*; ; *DIP-γ^-/-* 50.5+ /- 7.8 (n = 6), ****p<0.0001; *lGMR Gal4* control 333+ /- 9.8 (n = 6), *lGMR Gal4 >UAS-dpr11* 509.7+ /- 41.5 (n = 6), ****p<0.0001. (D) pDm8 cell numbers decrease dramatically when R7 PRs are absent, when they are converted to the yR7 fate (*ss^GOF*), or when they all express Dpr11 (*dpr11^GOF*). pDm8 cell number in wild-type, *ss^GOF*, *sev* mutant, Dpr11 overexpression: *DIP-γ^-/+* 179.4+ /- 25.6 (n = 11), *ss^GOF*, *DIP-γ^-/+* 28.7+ /- 4.3 (n = 6), ****p<0.0001; *ss^GOF*, *DIP-γ^-/+* 28.7+ /- 4.3 (n = 6), *ss^GOF*, *DIP-γ^-/-*59.5+ /- 8.8 (n = 6), ****p<0.0001; *DIP-γ^-/-*204.9+ /- 27.2 (n = 12), *ss^GOF*, *DIP-γ^-/-*59.5+ /- 8.8 (n = 6), ****p<0.0001; *lGMR Gal4* control 127.7+ /- 18.7 (n = 6), *lGMR Gal4 >UAS-dpr11* 42.2+ /- 5.6 (n = 6), ****p<0.0001; *sev^-/+*; ; *DIP-γ^-/+* 188.5+ /- 10.2 (n = 6), *sev^-/-*; ; *DIP-γ^-/+* 39.3+ /- 6.3 (n = 6), ****p<0.0001; *sev^-/-*; ; *DIP-γ^-/+* 39.3+ /- 6.3 (n = 6), *sev^-/-*; ; *DIP-γ^-/-* 52.5+ /- 4.8 (n = 6), **p=0.002. (E-F, H) Dpr11

*Figure 9 continued on next page*

*Figure 9 continued*

overexpression in the retina converts all medulla columns to y by selecting for yDm8. Pupal medullary neuropil (~45 hr - 48hr APF) of (E) *lGMR-Gal4* control and (F) *lGMR-Gal4 >UAS-dpr11* labeled with anti-Chp (red) and anti-DIP-γ (green). Cross-section views of the medulla shown (E-F). Two p columns (red only) are circled in (E). Quantitation in (H). Maximum intensity projection; scale bar 5 μm. (G) Conversion of all R7 PRs to pR7 fate results in loss of yDm8 neurons. yDm8 cell number in *dve^GOF* counted with anti-Dac and anti-GFP for *DIP-γ^{Mi>GFP}* reporter. *lGMR Gal4* control 271 +/- 18.3 (n = 6), *lGMR Gal4 >UAS* Dve 110.3+ /- 3.8 (n = 6), ****p<0.0001. Graph shows mean + /- std. deviation and unpaired Student's t-test p-values. (H) Percentage of Dm8 home columns that are yDm8 (quantitated from images like those in E and F). *lGMR Gal4/+* 0.62+ /- 0.02 (n = 199, 5 OLs), *lGMR Gal4 >UAS-dpr11* 0.98+ /- 0.01 (n = 234, 6 OLs), ****p<0.0001; n represents total number of columns analyzed. Graph shows mean + /- std. deviation and unpaired Student's t-test p-values.

The online version of this article includes the following figure supplement(s) for figure 9:

**Figure supplement 1.** Changing R7 fate or expressing Dpr11 affects yDm8 and pDm8 survival.
**Figure supplement 2.** yDm8 arbors do not innervate pR7 home columns in *dpr11* mutants.

To complete our analysis of the consequences of changing R7 fates, we examined ectopic expression of Dpr11 in the eye (*dpr11^GOF*). We reasoned that since *dpr11* was expressed in all R7 in *ss^GOF* (*Figure 9B*), Dpr11 overexpression in the eye should mimic *ss^GOF*. Indeed, the M6 layer showed continuous yDm8 labeling with no pDm8 gaps, similar to *ss^GOF* (*Figure 9—figure supplement 1D*). This was confirmed by determining yDm8 and pDm8 cell numbers in *dpr11^GOF*. The yDm8 population increased by 53% over control when Dpr11 was overexpressed in all PRs, accompanied by a 67% loss of pDm8 (*Figure 9C–D*). Dpr11 overexpression in *dpr11^GOF* was confirmed by labeling with Dpr11 antibody. Dpr11 overexpression does not convert pR7 to the yR7 fate, because R7 PRs that ectopically express Dpr11 do not all express Rh4 (*Figure 9—figure supplement 1G–I*).

We next examined cross-sectional (top-down) views of the mid-pupal medulla in *dpr11^GOF* labeled with DIP-γ and Chp antibodies to assess the composition of Dm8 columns when Dpr11 is expressed in all R7 PRs. Wild-type controls showed 62y:38p columns in the medulla, similar to the 65y:35p ratio of yellow and pale ommatidia in the eye. By contrast, in *dpr11^GOF* only yellow columns were observed, indicating that all pDm8 home columns had been replaced by yDm8 (*Figure 9E–F*). Thus, ~35% of yDm8 neurons had now selected PRs that express Dpr11 but were otherwise of the p subtype as their home column R7 (*Figure 9H*, *Figure 9—figure supplement 1I*). Similar results were obtained in *ss^GOF*, in which pR7 PRs are actually converted to the y subtype (*Figure 9—figure supplement 1J*). Interestingly, we did not observe any mistargeting of yDm8 to the M3 layer in *dpr11^GOF*, although Dpr11 was overexpressed in all PRs including R8 (*Figure 9—figure supplement 1L*).

Since ectopic expression of Dpr11 in all PRs changed the identities of home column Dm8, converting them all to yDm8 (*Figure 9E–F,H*), we investigated whether there were defects in home column selection in *dpr11* LOF mutants. We examined mutants and heterozygous controls at 44 hr APF because Dm8 have not yet contacted neighboring columns at this stage, making it possible to unambiguously assign the home column (*Ting et al., 2014*). y and p R7 were identified using *dpr11^{Mi>GFP}* and Chp antibody. Using the DIP-γ antibody to label yDm8, we determined how many yR7 containing columns were paired with yDm8. In controls, every yR7 column had DIP-γ labeling adjacent to it. In *dpr11* mutants, there was a significantly higher percentage of unpaired yR7 as compared to controls (*Figure 9—figure supplement 2D*). However, this result can be explained by the fact that yDm8 die in *dpr11* mutants, so that the number of surviving yDm8 neurons is insufficient to innervate all yR7. Thus, the absence of DIP-γ labeling adjacent to yR7 does not indicate that Dpr11 is required for home column selection. We found no mistargeting errors where pR7 containing columns labeled only with Chp were apposed to yDm8 in *dpr11* mutants (*Figure 9—figure supplement 2A–B*).

We repeated the above experimental paradigm in a genotype where yDm8 cell death was inhibited in a *dpr11* mutant and found that this eliminated unpaired yR7. We observed very rare mistargeting of a yDm8 to a pR7 in this genotype (2/119 pR7 (or 300 total R7 PRs) examined) (*Figure 9—figure supplement 2C*). We hypothesize that pR7 may not be available as yDm8 targets in *dpr11* mutants because they would have been selected by pDm8 neurons. The fact that pR7 are occupied by pDm8 makes it difficult to assess whether Dpr11-DIP-γ interactions are required for selection of yR7 as targets by yDm8.

## Discussion

A circuit for UV wavelength discrimination in *Drosophila* is defined by expression of a pair of cell-surface IgSF binding partners, Dpr11 and DIP-γ. Dpr11 is selectively expressed by Rh4[+] yR7 PRs (*Figure 1*), which connect to the yDm8 subtype of Dm8 amacrine neurons that express DIP-γ. Rh3[+] pR7 connect to the pDm8 subtype, which is DIP-γ negative (*Figure 2*). yR7 and yDm8 synapse onto Tm5a neurons, which also express DIP-γ and project to the lobula, while pR7 and pDm8 synapse onto Tm5b (*Figures 3* and *4* and associated videos). Dpr11-DIP-γ interactions are also involved in determination of Dm8 arbor morphology, which we analyzed using ExM (*Figures 5* and *6* and associated videos). yDm8 neurons are generated in excess during development and compete for yR7 partners. Their survival is controlled by signaling mechanisms that require interaction between DIP-γ on the yDm8 neurons and Dpr11 on their presynaptic home column yR7 PRs (*Figures 7* and *8*). yDm8 and pDm8 neurons do not normally compete for survival signals, but can be forced to do so by changing R7 subtype fate (*Figure 9*).

### Control of yDm8 survival and R7-Dm8 connectivity by Dpr11-DIP-γ signaling

Yellow and pale ommatidia are generated by a stochastic process and are distributed in a ~ 65y:35p ratio in the retina (*Wernet et al., 2006*). Since yDm8 and pDm8 are separate populations (*Figure 2*) and R7 y vs. p fates are randomly determined, what mechanisms ensure that each yDm8 has a yR7 partner and each pDm8 has a pR7 partner? The strategy appears similar to those used in the development of many mammalian nervous system circuits, in that yDm8 neurons are generated in excess of their final cell numbers in adults, and those that are not selected for survival by an appropriate home column R7 partner die through apoptosis.

Our conclusions are summarized in a model (*Figure 10*). There are two subtypes of Dm8: yDm8 that express DIP-γ and pDm8 that do not. yDm8 are born in excess of the final cell numbers that are present in the adult (*Figure 8A*). There are likely to be extra pDm8 born as well, but due to the lack of a specific pDm8 marker we cannot determine their cell numbers in early pupae. Dpr11, the binding partner of DIP-γ, is expressed exclusively in yR7 in the retina (*Figure 1E–F*), allowing yDm8 neurons to select yR7 PRs as their appropriate partners. The extra yDm8 that do not find a yR7 partner undergo cell death due to loss of survival signals mediated by DIP-γ-Dpr11 interactions. This selection mechanism ensures that the ratio of yDm8 to pDm8 in the medulla matches the ratio of yR7 to pR7 in the eye (*Figure 1I*).

Dpr11 and DIP-γ are required for survival, because 50–60% of yDm8 die when either molecule is absent (*Figures 7*, *8A* and *10*). If all R7 are converted to yR7 (in ss^GOF), or if they all express Dpr11, many more yDm8 neurons survive than in wild-type, showing that the numbers of Dpr11-expressing R7 are limiting for yDm8 survival in wild-type animals (*Figures 9C* and *10*). This seems surprising, because a wild-type retina should contain 460–500 yR7 (*Posnien et al., 2012*), which is greater than the number of yDm8 we count in early pupae (*Figure 8A*). However, the actual yDm8 cell numbers may be larger due to incomplete penetrance of the markers, as discussed in Results. yDm8 and pDm8 populations do not compete with each other in wild-type because they have different R7 subtype partners. When the numbers of yDm8 neurons are increased by suppressing apoptotic cell death with DIAP in yDm8, pDm8 cell numbers do not decrease (*Figure 7B*). However, yDm8 and pDm8 neurons can be forced to compete by changing R7 subtype fate. When all R7 PRs are converted to yR7, or when they all express Dpr11, almost all pDm8 neurons die (*Figures 9D* and *10*). Conversely, most yDm8 neurons die when all R7 PRs are converted to pR7 (*Figure 9G*). pDm8 neurons are preferentially selected by pR7 PRs, using unknown molecular mechanisms, and pR7 PRs are needed to ensure pDm8 survival. This is demonstrated by the fact that pDm8 are unable to effectively fill vacant yR7 slots. For example, in ss^GOF, DIP-γ animals, there are at least 180 yR7 slots that are made vacant by loss of yDm8, but the number of pDm8 neurons only increases by ~30 (~2 fold) in this genotype (*Figure 9C–D*).

### Selection of yDm8 neurons for survival occurs early, prior to synaptogenesis

Dpr11 is already expressed in a subset of R7 PRs at 15 hr APF, indicating that yR7 fate is determined by that time (*Figure 8F*). DIP-γ is expressed by 18 hr - 20 hr APF in select Dm8 neurons apposed to

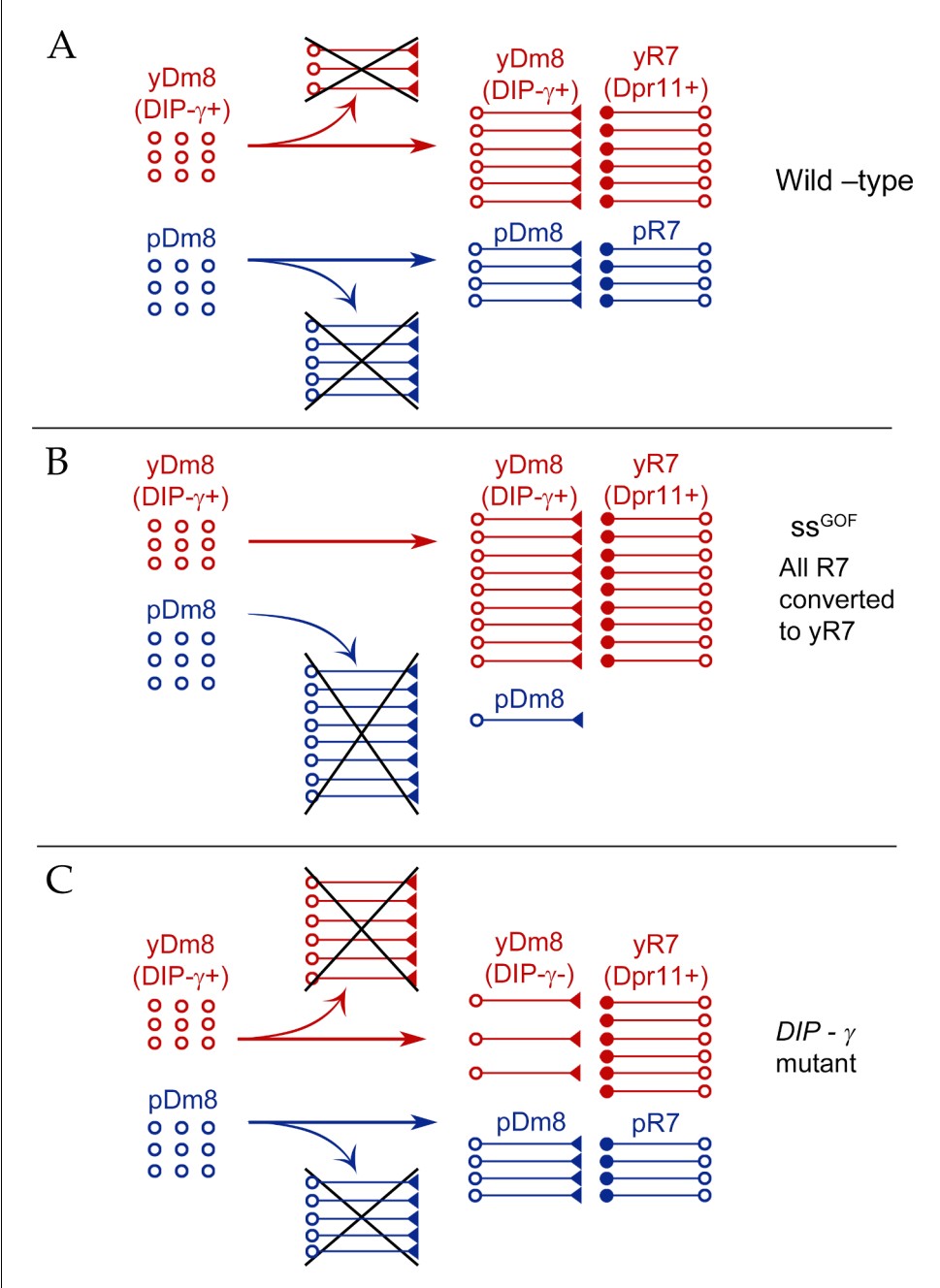

**Figure 10.** Model for Dm8 selection and survival. (**A**) In wild-type, yDm8 neurons (red) are produced in excess and are selected by yR7 PRs for connectivity and survival. pDm8 neurons (blue) are also shown as being generated in excess. We have indicated their cell numbers prior to selection as being the same as yDm8, but this is arbitrary, since we do not know how many pDm8 are born. Unselected yDm8 and pDm8 neurons die due to the absence of survival signals (indicated by Xs).~30% of the yDm8 neurons present at 15 hr APF die in wild-type (*Figure 8A*). (**B**) When all R7 PRs are converted to yR7 in ss$^{GOF}$ (or when they all express Dpr11), more yDm8 neurons survive and almost all pDm8 neurons die (*Figure 9*). (**C**) In *DIP-γ* mutants, yDm8 neurons are not selected by yR7 PRs and more of them die. ~75% of the yDm8 neurons present at 15 hr APF die in *DIP-γ* mutants (*Figure 8A*). Similar results are observed for *dpr11* mutants. This means that some yR7 PRs remain uninnervated (*Figure 9—figure supplement 2*).

Dpr11-expressing R7 PRs (*Figure 8G*; *Figure 8—figure supplement 1A–B*). At this time, R7 growth cones are positioned in the incipient layer (*Kulkarni et al., 2016*; *Ting et al., 2005*). This is consistent with earlier findings that single-cell Dm8 clones contact home column R7 growth cones by 17 hr APF (*Ting et al., 2014*). Taken together, these results indicate that yR7 and yDm8 have met and selected each other in the incipient layer by 15 hr - 20 hr APF. Survival signaling mediated by Dpr11-DIP-γ

interactions is probably initiated around that time, because cell death of yDm8 in *DIP-γ* mutants begins to occur between 15 hr and 25 hr APF (*Figure 8A*).

There is widespread cell death in the medullary cortex as part of normal optic lobe development. This occurs in two phases, with the first phase spanning 0 hr - 48 hr APF and peaking at 24 hr APF (*Togane et al., 2012*). In wild-type, ~30% of the yDm8 neurons present at 15 hr APF die by 45 hr APF, at which time yDm8 cell numbers are the same as those in adult. In *DIP-γ* mutants, ~75% are lost by 45 hr APF. If apoptotic cell death is suppressed by DIAP, then the yDm8 population in adult remains the same as at 15 hr APF (*Figure 8A*). We suggest that the yDm8 neurons that die in wild-type are those that are in excess of the number that can be selected for survival by Dpr11-expressing yR7 PRs. Since most yDm8 neurons require DIP-γ for survival, we infer that the excess yDm8 neurons in wild-type die because they lack survival signals that are transmitted through Dpr11-DIP-γ interactions. These selection events are independent of synaptogenesis between R7 and Dm8, which occurs only after 60 hr APF.

The DIP-α-expressing Dm4 neurons are also generated in excess, and their survival is regulated by interactions between DIP-α and its partners Dprs 6 and 10, which are expressed on presynaptic L3 neurons. In *DIP-α* or *dpr6 dpr10* mutants, about 50% of Dm4 and 20% of Dm12 neurons are lost. If cell death is blocked in wild-type, the cell numbers of Dm4 neurons in adults are increased by about 25% (*Xu et al., 2018*). Thus, in these cases, Dpr-DIP interactions play a similar role in suppressing cell death of postsynaptic neurons in the distal medulla. However, not all DIP-γ expressing neurons in the optic lobe require it for survival. A subset of lobula plate tangential cells (LPTCs) also express DIP-γ, and these do not die in *DIP-γ* mutants (unpublished data).

Our gain-of-function studies show that when Dpr11 is expressed in all R7, all pDm8 home columns are converted to yDm8 (*Figure 9E–F,H*), demonstrating that expression of Dpr11 in R7 PRs that are otherwise of the pale subtype is sufficient for selection of yDm8. However, we were unable to determine whether Dpr11 is also necessary for yDm8-yR7 pairing by analysis of LOF mutants. We did not observe mistargeting of surviving yDm8 in *dpr11* mutants to pR7 columns, most likely because pR7 slots are occupied by pDm8 neurons (*Figure 9—figure supplement 2A–D*). Another factor that could contribute to accurate targeting in *dpr11* mutants is that there may be a second mechanism for support (and possibly selection) of yDm8 by R7 that is independent of Dpr11-DIP-γ interactions. This is suggested by the fact that ~70% of yDm8 are lost in *sev* mutants, but only ~50% in *DIP-γ* and *dpr11* mutants (*Figures 9C* and *7A*).

## Effects of *DIP-γ* and *dpr11* mutations on yDm8 arbor morphology

A typical wild-type yDm8 arbor has a thick distal dendritic projection (sprig) that extends along the home column yR7 terminal, usually reaching M4. In *DIP-γ* and *dpr11* mutants, the distribution of sprig diameters for yDm8 neurons is shifted toward smaller values (*Figure 5I*), suggesting that mutant sprigs may have a reduced capacity to form synapses with yR7 in M4 and M5. Synapses that would have been in M4 and M5 might then be found in the base of the arbor in M6. EM reconstruction data for columns D and E supports this idea. R7 output synapses (T-bars) are mainly in M4 and M5 (on the sprig) in column E, which has a Dm8 with a thick sprig (*Figures 2A* and *4A*, inset in 4B). In column D, whose Dm8 has a very thin sprig, R7 T-bars are located in M6 (in the main arbor) (*Figure 4A*, inset in 4C).

The effects of Dpr11 and DIP-γ on yDm8 sprig morphology may involve a different pathway from that involved in cell death. Although cell death would have occurred in the *DIP-γ* and *dpr11* mutant animals, these flipouts were analyzed in adults, in which the population has stabilized and the remaining yDm8 neurons are not in the process of dying. Sprig morphology defects might be a consequence of a loss of Dpr11-DIP-γ-mediated adhesion between yR7 and yDm8, since *dpr11* and *DIP-γ* mutants are similarly affected.

We had earlier reported an 'overshoot' phenotype in *dpr11*[Mi>GFP]/Df mutants in which some R7 terminals had processes extending beyond M6 into proximal medulla layers (*Carrillo et al., 2015*). Overshoots were labeled by an overexpressed truncated form (Brp-short) of the active zone protein Bruchpilot (Brp) (*Berger-Müller et al., 2013*). However, this phenotype was not observed in the *dpr11*[CRISPR] null mutant when endogenous Brp puncta were labeled with the STaR method (*Xu et al., 2018*). To determine whether this discrepancy was due to the *dpr11*[Mi>GFP]/Df genotype or to the Brp-short marker, we analyzed *dpr11*[CRISPR] homozygotes using Brp-short and observed a significant increase in yR7 overshoots as compared with the heterozygous control (*Figure 9—figure*

*supplement 2E*). However, the interpretation of this Brp-short phenotype is unclear, as it does not correspond to a shift in the distribution of endogenous Brp puncta into layers proximal to M6.

## A circuit for wavelength discrimination defined by Dpr11 and DIP-γ expression

Tm5a, Tm5b, and Tm5c are output neurons for R7 and R8 circuits. Their dendrites receive direct synaptic input from R7, R8, and Dm8, and their axons relay signals to the lobula (*Gao et al., 2008*; *Karuppudurai et al., 2014*; *Takemura et al., 2013*; *Takemura et al., 2015*). It has been hypothesized that Tm5a/b/c cells are analogous to vertebrate retinal ganglion cells, and that true color vision (intensity-independent hue discrimination) involves interpretation of Tm inputs by circuits in the lobula. Chi-Hon Lee and his colleagues developed a learning paradigm in which flies discriminate

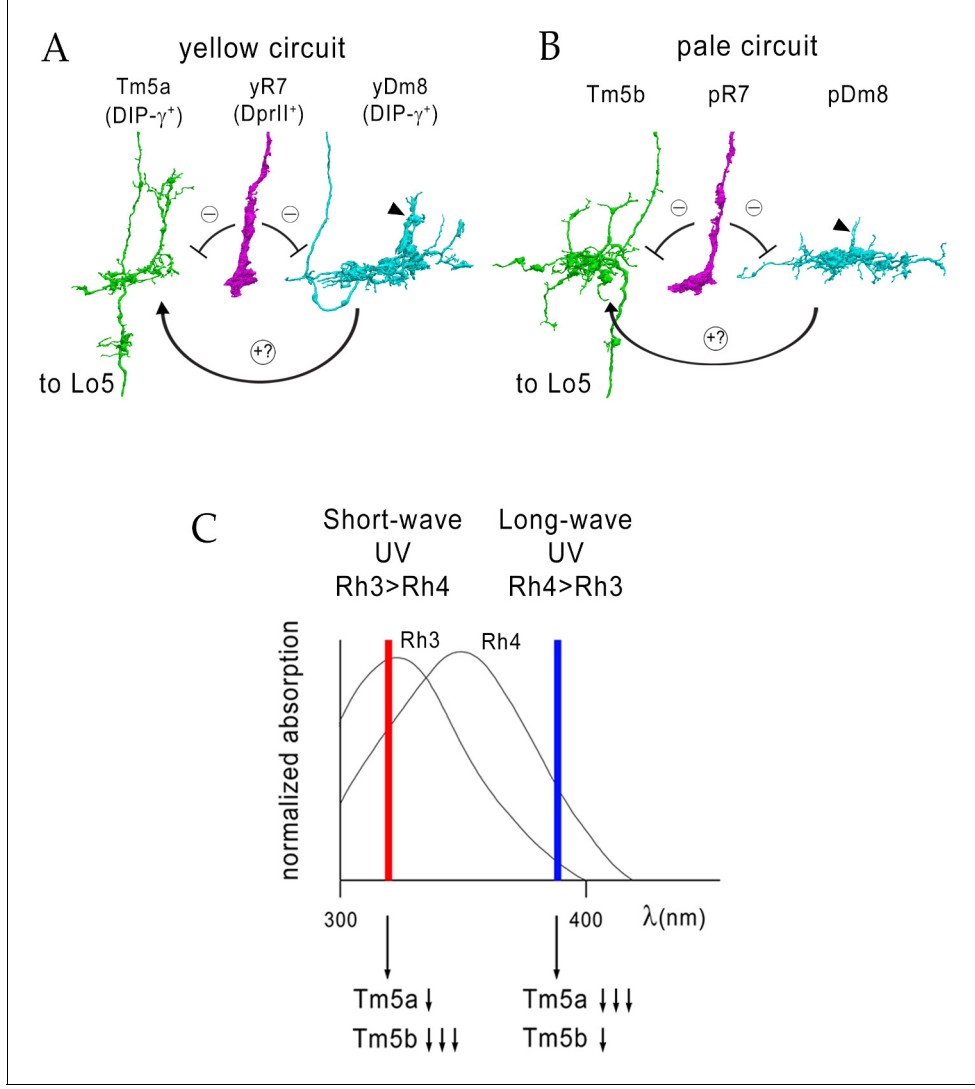

**Figure 11.** Model for UV wavelength discrimination by yellow and pale circuits. (**A**) Cells in a yellow circuit, from column E. yR7-E, yDm8-E, and Tm5a-E are shown. yR7 inhibits both yDm8 and Tm5a (repression bars). yDm8 makes glutamatergic synapses (probably excitatory) onto Tm5a (arrow). Tm5a and Tm5b axons project to the 5th layer of the lobula. (**B**) Cells in a pale circuit, from column D. pR7-D, pDm8-D, and Tm5b-D are shown. The connections among the cells are the same as for the yellow circuit. The implication of these connection patterns is that R7 stimulation might inhibit Tm5a/b by both direct (histaminergic) and indirect (histaminergic inhibition of excitatory Dm8 glutamatergic output) pathways. yDm8-E (in **A**) and pDm8-D (in **B**) sprigs indicated by arrowheads. (**C**) Short-wave UV (red bar) would stimulate Rh3+ pR7 more than Rh4+ yR7, and might therefore produce more inhibition of Tm5b than of Tm5a. Long-wave UV (blue bar) would produce more inhibition of Tm5a than of Tm5b. These signals could be read out by Lo neurons that receive Tm5a and Tm5b inputs.

between equiluminant blue and green light, which are differentially sensed by y (Rh6) and p (Rh5) R8. Four classes of Tm neurons (Tm5 a/b/c and Tm20) make redundant contributions to this response (*Melnattur et al., 2014*).

The absorption spectra of Rh4 (yR7) and Rh3 (pR7) extensively overlap (*Figure 11C*), and it is unlikely that a learned color vision paradigm could be developed that would distinguish between Rh4 and Rh3 inputs. Nevertheless, the existence of discrete y and p R7 channels suggests that the fly utilizes them to make discriminations among UV wavelength inputs. To make such discriminations would require that the y and p channels have different synaptic circuits with distinguishable outputs. Our analysis of the patterns of synaptic connections defined by the EM reconstruction of the medulla (*Takemura et al., 2013*; *Takemura et al., 2015*) indicates that yR7 and yDm8 both preferentially synapse onto Tm5a, while pR7 and pDm8 synapse onto Tm5b (*Figure 3*). Tm5a expresses DIP-γ, while Tm5b does not (*Cosmanescu et al., 2018*), so Dpr11-DIP-γ interactions might be involved in specifying both yR7-yDm8 and yR7-Tm5a connections. Four of the seven Dm8 neurons, and three of the eight R7 PRs, also make a few synapses onto Tm5c neurons, which are required for innate UV preference (*Karuppudurai et al., 2014*) (*Figure 3—source data 1*). There is no specificity for y *vs.* p in the connections to Tm5c.

We visualized yR7-yDm8-Tm5a (yellow) and pR7-pDm8-Tm5b (pale) circuits using 3D renderings of cells from the EM reconstruction (*Takemura et al., 2015*) (*Figure 4A–C* and associated videos). The yDm8-E sprig and Tm5a-E dendritic branch are both wrapped tightly around the yR7-E terminal, and most yR7 T-bars are apposed to the sprig and branch within layer M5. Dm8 arbors contain both pre- and postsynaptic elements, and therefore both receive inputs and emit outputs. yDm8-E T-bars are distributed between the sprig and the main arbor in M6 (*Figure 4B*). Remarkably, our 3D renderings of wild-type yDm8 and yR7 neurons from ExM (*Figure 6* and associated videos) are almost superimposable on yDm8-E and yR7-E from the EM reconstruction (*Figure 6—figure supplement 1*). This allows us to interpret yDm8 ExM phenotypes in *DIP-γ* mutants (*Figure 6*, *Figure 6—figure supplement 2*) by reference to the EM reconstruction.

R7 is histaminergic, and inhibits both Dm8 and Tm5a/b (*Figure 11A–B*), usually through polyadic synapses where a single R7 T-bar is apposed to both a Dm8 and a Tm5a/b postysnapse (*Figure 4D–E*). Dm8 is glutamatergic, while Tm5a and Tm5b are cholinergic (*Davis et al., 2018*; *Gao et al., 2008*; *Karuppudurai et al., 2014*). This suggests that yR7 input might cause inhibition of Tm5a in two ways: by direct inhibition and by inhibiting the glutamatergic input (probably excitatory) of yDm8 onto Tm5a. Conversely, pR7 input could preferentially inhibit Tm5b by both direct and indirect mechanisms. There might be timing differences between direct and indirect inhibition.

UV inputs will always stimulate both y and pR7 channels, because the Rh4 and Rh3 absorption maxima differ by only 20 nm. However, longer-wave inputs that activate Rh4 on yR7 more than Rh3 on pR7 might cause more inhibition of Tm5a than of Tm5b, while the reverse would be true of shorter-wave inputs that preferentially activate Rh3 on pR7 (*Figure 11*). Combining direct (R7→Tm5a/b) and indirect (R7→Dm8→Tm5a/b) inhibition of Tm5a and Tm5b outputs might amplify these effects, depending on the relative timing of R7 and Dm8 inputs. This model suggests that neurons in the lobula or elsewhere that can read the ratio of Tm5a to Tm5b output mediate UV wavelength discrimination. Tm5a/b/c and Tm20 as a group synapse onto many lobula neuron types, but the specific partners of Tm5a and Tm5b are mostly unknown. Interestingly, however, all Tm5a but only half of Tm5b neurons synapse onto LT11 lobula projection neurons (*Lin et al., 2016*). Flies with silenced LT11 neurons have reduced phototaxis toward blue light (*Otsuna et al., 2014*).

In conclusion, our results show that Dpr11 and DIP-γ expression patterns define a yR7-yDm8-Tm5a color vision circuit that should preferentially respond to longer-wavelength UV input. Suppression of yDm8 cell death triggered by engagement of Dpr11 on yR7 by DIP-γ on yDm8 helps to build this circuit by ensuring that each yDm8 has a home column yR7.

## Materials and methods

### *Drosophila* genetics

Genotypes of all the flies analyzed in this study are indicated in *Tables 2–3* and lines obtained from other sources are listed in *Table 1*. Heterozygote controls were used in all experiments for determining cell numbers. *DIP-γ^{Mi>GFP}* reporter (indicated in all graphs as *DIP-γ ^{-/+}*) is an insertion in the 5'

**Table 1.** Lines and sources.

| Genotype | Source |
| --- | --- |
| yw hsFlp; UAS > CD2, y+>mCD8::GFP/Cyo; TM2/TM6B | Gift from C-H Lee |
| Rh4-lacZ | Bloomington |
| R24F06-p65.AD | Bloomington |
| DIP-γ Gal4-DBD | Gift from C Desplan |
| OK371-VP16AD | Gift from C-H Lee |
| R24F06-Gal4 | Bloomington, (*Nern et al., 2015*) |
| DIP-γ$^{MI03222}$(DIP-γ$^{Mi>GFP}$), DIP-γ$^{MI03222}$ Gal4 | HJ Bellen |
| DIP-γ$^{null}$ | Gift from L Zipursky (*Xu et al., 2018*) |
| dpr11$^{MI02231-GFP}$ (dpr11$^{Mi>GFP}$) | HJ Bellen |
| dpr11$^{null}$ | Gift from L Zipursky (*Xu et al., 2018*) |
| dpr11 RNAi GD23243 (III) | VDRC |
| UAS-DIP-γ$^{sh}$ (II) | This study |
| UASp-DIAP I (II) | Bloomington |
| UAS-DIAP1-myc/TM6b | Gift from L Zipursky (*Xu et al., 2018*) |
| UAS-dveA-9B2/TM3 | Gift from Hideki Nakagoshi (*Nakagawa et al., 2011*) |
| lGMR-Ss (II) | Gift from C Desplan |
| lGMR-Gal4 | Bloomington |
| sev$^{14}$ | Bloomington |
| UAS-dpr11$^{sh}$ (II) | This study |
| Rh4-lexA::p65, lexAop2-brp-short$^{cherry}$ | Gift from T Suzuki |
| UAS-DIP-α-V5 | Gift from L Zipursky (*Xu et al., 2018*) |

UTR intron and has reduced or no protein expression. Thus, this line serves as a mutant as well as a reporter of *DIP-γ* transcript.

For eye-specific transgenic RNAi, we screened three *dpr11* RNAi lines by crossing to a line which had one copy of *dpr11* removed to increase the effectiveness of the RNAi line: *lGMR-Gal4; dpr11$^{null}$, DIP-γ$^{MI03222-GFP}$*; the RNAi line GD2343 (VDRC) had the strongest phenotype and was used in the paper.

Generation of UAS-transgenic flies cDNA encoding Dpr11 and DIP-γ were cloned from pOT2 GH22307 for Dpr11 and pOT2 GH08175 for DIP-γ into pUAST attB vector using standard molecular biology techniques. 5' UTRs for both genes contained many upstream ATG codons; we made deletions of the 5' UTRs so that the ATGs of the proteins were the first ATGs in the mRNAs, in order to increase expression of the transgenes (sequences available on request). Transgenes were injected into embryos (Rainbow Transgenics). UAS-Dpr11 and UAS-DIP-γ both used the attP40 (2L) landing site.

## Antibodies

The primary antibodies used were as follows: anti-DIP-γ (guinea pig, 1:200) and anti-Dpr11 (rabbit, 1:150) were gifts from C. Desplan. Anti-Pros MR1A (mouse 1:4), anti-Elav 7E8A10 (rat, 1:10), anti-Dac 2–3 (mouse 1:50), anti-chaoptin 24B10 (mouse 1:20) were obtained from Developmental Studies Hybridoma Bank (University of Iowa, IA). Commercial antibodies were used as follows: Rabbit anti-RFP (Rockland, 1:500), rabbit anti-GFP (Thermo- Fisher Scientific, 1:500), chicken anti-GFP (Aves labs, 1:500), mouse anti-myc 9E10 (Abcam, 1:500) and chicken anti-beta-galactosidase (Abcam, 1:1000). Secondary antibodies were obtained from Thermo-Fisher Scientific and used at 1:500.

**Table 2.** List of genotypes in figures and graphs.

| Figures | Short genotype | Complete genotype |
|---|---|---|
| 1E | $dpr11^{Mi>GFP}/+$ | $dpr11^{MI02231-GFP}/+$ |
| 1 F-F' | | $dpr11^{MI02231-GFP}/Rh4-LacZ$ |
| 1 G-G', H | | $10xUAS-mCD8::RFP/+; DIP-\gamma^{MI03222}/R24F06-Gal4$ |
| 1I | | $dpr11^{MI02231-GFP}/Rh4-LacZ$ and $10xUAS-mCD8-RFP/+; DIP-\gamma^{MI03222}/R24F06-Gal4$ |
| 2B, D | | $yw\ hsflp/+; UAS > CD2, y+>mCD8::GFP/R24F06-p65.AD; Rh4-lacZ/DIP-\gamma^{MI03222}Gal4.DBD$ |
| 2C, E | | $yw\ hsflp/+; UAS > CD2, y+>mCD8::GFP/+; Rh4LacZ/R24F06-Gal4$ |
| 2F | $dpr11^{Mi>GFP}/+$ | $dpr11^{MI02231-GFP}/+$ |
| 2G | DIP-γ Gal4 > Flp, Pan-Dm8 LexA > FSF GFP | $20xUAS-flp/+; R24F06-LexA/+; DIP-\gamma^{MI03222}Gal4, LexAop\ FRT > stop > FRT\ mCD8::GFP/+$ |
| 2 H-H' | $DIP-\gamma^{Mi>GFP}$, Pan-Dm8 > RFP | $10xUAS\ mCD8::RFP/+; DIP-\gamma^{MI03222-\ GFP}/R24F06-Gal4$ |
| 2I-J | yDm8 | $yw-hsflp/+; UAS > CD2,y+>mCD8GFP/R24F06-p65.AD; Rh4LacZ/DIP-\gamma\ Gal4\ DBD$ |
| 2I-J | pDm8 | $yw-hsflp/+; UAS > CD2,y+>mCD8\ GFP/Sp; R24F06Gal4, DIP-\gamma\ Gal80/Rh4LacZ$ |
| 2I-J | pDm8 | $yw-hsflp/+; UAS > CD2,y+>mCD8GFP/+; Rh4LacZ/R24\ F06Gal4$ |
| 5A-B, I, K | $DIP-\gamma^{-/+}$ | $yw\ hsflp/+; UAS > CD2, y+>mCD8::GFP/R24F06-p65.AD; Rh4-lacZ/DIP-\gamma^{MI03222}Gal4.DBD$ |
| 5C-D, I, K | $DIP-\gamma^{-/-}$ | $yw\ hsflp/+; UAS > CD2, y+>mCD8::GFP/R24F06-p65.AD; DIP-\gamma^{null}/DIP-\gamma^{MI03222}Gal4.DBD$ |
| 5E-F, I, K | $dpr11^{-/-}$ | $yw\ hsflp/+; UAS > CD2, y+>mCD8::GFP/Rh4LacZ; R24F06-Gal4, dpr11^{null}/dpr11^{null}$ |
| 5G, H | $DIP-\gamma^{-/+}$ | $yw-hsflp/+; UAS > CD2,y+>mCD8GFP/R24F06p65AD; Rh4LacZ/DIP-\gamma\ Gal4DBD$ |
| 5G | $DIP-\gamma^{-/-}$ | $yw\ hsflp/+; UAS > CD2, y+>mCD8::GFP/R24F06-p65.AD; DIP-\gamma^{null}/DIP-\gamma^{MI03222}Gal4.DBD$ |
| 5J, L | $DIP-\gamma^{-/+}$ | $yw-hsflp/+; UAS > CD2,y+>mCD8GFP/R24F06p65AD; Rh4LacZ/DIP-\gamma\ Gal4\ DBD$ |
| 5J, L | $DIP-\gamma^{-/-}$ | $yw-hsflp/+; UAS > CD2,y+>mcd8GFP/R24F06p65AD; DIP-\gamma\ Gal4\ DBD, Rh4LacZ/DIP-\gamma^{null}$ |
| 5J, L | $dpr11^{-/-}$ | $yw-hsflp/+; UAS > CD2,y+>mcd8GFP/Rh4LacZ; R24F06Gal4, dpr11^{null}/dpr11^{null}$ |
| 6A-C and *Figure 6—videos 1–2* | WT | $yw\ hsflp/+; UAS > CD2, y+>mCD8::GFP/R24F06-p65.AD; Rh4-lacZ/DIP-\gamma^{MI03222}Gal4.DBD$ |
| 6D and *Figure 6—videos 3–4* | $DIP-\gamma^{-/-}$ | $yw\ hsflp/+; UAS > CD2, y+>mCD8::GFP/R24F06-p65.AD; DIP-\gamma^{null}/DIP-\gamma^{MI03222}Gal4.DBD$ |
| 7A-B, D | $DIP-\gamma^{-/+}$ | $10xUAS-mCD8::RFP/+; R24F06-Gal4/DIP-\gamma^{MI03222-GFP}$ |
| 7A-B | $DIP-\gamma^{-/-}$ | $10xUAS-mCD8::RFP/+; R24F06-Gal4, DIP-\gamma^{null}/DIP-\gamma^{MI03222-GFP}$ |
| 7A-B | $dpr11^{-/+}$ control | $10xUAS-mCD8::RFP/+; R24F06-Gal4, dpr11^{null}/DIP-\gamma^{MI03222-GFP}$ |
| 7A-B, H | $dpr11^{-/-}$ | $10xUAS-mCD8::RFP/+; R24F06-Gal4, dpr11^{null}/dpr11^{null}, DIP-\gamma^{MI03222-GFP}$ |
| 7A-B | $dpr11^{-/-}, DIP-\gamma^{-/-}$ | $10xUAS-mCD8::RFP/+; R24F06-Gal4, dpr11^{null}, DIP-\gamma^{null}/dpr11^{null}, DIP-\gamma^{MI03222-GFP}$ |
| 7A | $DIP-\gamma^{-/+}$ control | $R24F06-LexA/+; 13xLexAop-tdTomato::myr, DIP-\gamma^{MI03222-GFP}/+$ |
| 7A, E | $DIP-\gamma^{-/-}$ control | $R24F06-LexA/+; DIP-\gamma^{MI03222}\ Gal4/13xLexAop-tdTomato::myr, DIP-\gamma^{MI03222-GFP}$ |

*Table 2 continued on next page*

*Table 2 continued*

| Figures | Short genotype | Complete genotype |
| --- | --- | --- |
| 7A, F | *DIP-γ rescue* | *UAS-DIP-γ$^{sh}$/R24F06-LexA; DIP-γ$^{MI03222}$ Gal4/ 13xLexAop-tdTomato::myr, DIP-γ$^{MI03222-GFP}$* |
| 7A-B, G | yDm8 *DIAP rescue* | *UASp-DIAP/R24F06-LexA; DIP-γ$^{MI03222}$ Gal4/ 13xLexAop-tdTomato::myr, DIP-γ$^{MI03222-GFP}$* |
| 7C | *DIP-γ$^{-/+}$* | *DIP-γ$^{MI03222-GFP}$/+* |
| 7C | *DIP-γ$^{-/-}$* | *DIP-γ$^{MI03222-GFP}$/DIP-γ$^{null}$* |
| 7C | *dpr11$^{-/-}$* | *dpr11$^{null}$/dpr11$^{null}$, DIP-γ$^{MI03222-GFP}$* |
| 7C | *lGMR Gal4 control* | *lGMR-Gal4/+; dpr11$^{null}$, DIP-γ$^{MI03222-GFP}$/+ @29 deg* |
| 7C | *lGMR Gal4/dpr11 RNAi* | *lGMR-Gal4/+; dpr11$^{null}$, DIP-γ$^{MI03222-GFP}$/dpr11 RNAi GD23243 (III) @29 deg* |
| 7C | *lGMR Gal4/+* | *lGMR-Gal4/+; DIP-γ$^{MI03222-GFP}$/+* |
| 7C | *lGMR Gal4/UAS-DIP-γ* | *lGMR-Gal4/UAS-DIP-γ; DIP-γ$^{MI03222-GFP}$/+* |
| 7C | *lGMR Gal4/UAS-DIP-α* | *lGMR-Gal4/+; DIP-γ$^{MI03222-GFP}$/UAS-DIP-α-V5* |
| 8A, B-D | *DIP-γ$^{-/+}$* | *DIP-γ$^{MI03222-GFP}$/+* |
| 8A | *DIP-γ$^{-/-}$* | *DIP-γ$^{MI03222-GFP}$/DIP-γ$^{null}$* |
| 8A | yDm8 *DIAP rescue* | *UASp-DIAP/+; DIP-γ$^{MI03222}$Gal4/DIP-γ$^{MI03222-GFP}$* |
| 8C-E | *DIP-γ$^{-/+}$* | *OK371 VP16/UAS-Apo; DIP-γ Gal4 DBD/+* |
| 8C-E | *DIP-γ$^{-/-}$* | *OK371 VP16/UAS-Apo; DIP-γ Gal4 DBD/DIP-γ$^{null}$* |
| 8E | *dpr11$^{Mi>GFP}$* | *dpr11$^{MI02231-GFP}$* |
| 8F | yw | *WT control* |
| 9A | WT | *dpr11$^{MI02231-GFP}$/+* |
| 9B | *ss$^{GOF}$* | *lGMR-ss/+; dpr11$^{MI02231-GFP}$/+* |
| 9C-D | *DIP-γ$^{-/+}$* | *10xUAS-mCD8::RFP/+; R24F06-Gal4/DIP-γ$^{MI03222-GFP}$* |
| 9C-D | *LGMR > ss; DIP-γ$^{-/+}$* | *10xUAS-mCD8::RFP/lGMR ss; DIP-γ$^{MI03222-GFP}$/R24 F06-Gal4* |
| 9C-D | *LGMR > ss; DIP-γ$^{-/-}$* | *10xUAS-mCD8::RFP/lGMR-ss; DIP-γ$^{MI03222-GFP}$/R24F06-Gal4, DIP-γ$^{null}$* |
| 9C-D | *DIP-γ$^{-/-}$* | *10xUAS-mCD8::RFP/+; DIP-γ$^{MI03222-GFP}$/R24F06-Gal4, DIP-γ$^{null}$* |
| 9C-D | *sev$^{-/+}$;; DIP-γ$^{-/+}$* | *sev$^{14}$/+;; R24F06-Gal4/UAS-mCD8::RFP, DIP-γ$^{MI03222-GFP}$* |
| 9C-D | *sev$^{-/-}$;; DIP-γ$^{-/+}$* | *sev$^{14}$/sev$^{14}$;; R24F06-Gal4/UAS-mCD8::RFP, DIP-γ$^{MI03222-GFP}$* |
| 9C-D | *sev$^{-/-}$;; DIP-γ$^{-/-}$* | *sev$^{14}$/sev$^{14}$;; R24F06-Gal4, DIP-γ$^{null}$/UAS-mCD8::RFP, DIP-γ$^{MI03222-GFP}$* |
| 9C-D | *lGMR Gal4 control* | *R24F06-LexA/+; 13xLexAop-tdTomato::myr, DIP-γ$^{MI03222-GFP}$/lGMR Gal4* |
| 9C-D | *lGMR Gal4> UAS-dpr11* | *UAS-Dpr11-sh/R24 F06-LexA; 13xLexAop-tdTomato::myr, DIP-γ$^{MI03222-GFP}$/lGMR Gal4* |
| 9E | WT | *lGMR- Gal4/+* |
| 9F | *dpr11$^{GOF}$* | *lGMR-Gal4/UAS-dpr11$^{sh}$* |
| 9G | *lGMR Gal4 control* | *lGMR-Gal4/+; DIP-γ$^{MI03222-GFP}$/+* |
| 9G | *lGMR Gal4> UAS-Dve* | *lGMR-Gal4/+; DIP-γ$^{MI03222-GFP}$/UAS-dveA-9B2* |
| 9H | *lGMR Gal4* | *lGMR- Gal4/+* |
| 9H | *lGMR Gal4> UAS-dpr11* | *lGMR-Gal4/UAS-dpr11$^{sh}$* |

**Table 3.** List of genotypes in Supplementary figures and graphs.

| Supplementary figures | Short genotype | Complete genotype |
|---|---|---|
| *Figure 2—figure supplement 1A* | yDm8 split-Gal4 driver | *yw hsflp/+;*<br>*UAS > CD2, y+>mCD8::GFP/R24 F06-p65.AD;*<br>*Rh4-lacZ/DIP-γ$^{MI03222}$Gal4.DBD* |
| *Figure 2—figure supplement 1A* | Pan-Dm8 driver | *yw hsflp/+; UAS > CD2, y+>mCD8::GFP/+;*<br>*Rh4LacZ/R24F06-Gal4* |
| *Figure 2—figure supplement 1C–C'* | | *yw hsflp/+;*<br>*UAS > CD2, y+>mCD8::GFP/OK371 dVP16.AD;*<br>*Rh4 LacZ/DIP-γ$^{MI03222}$Gal4.DBD* |
| *Figure 5—source data 1* | WT:<br>Pan-Dm8 driver | *yw hsflp/+; UAS > CD2, y+>mCD8::GFP/+; Rh4LacZ/R24 F06-Gal4* |
| *Figure 5—source data 1* | WT: yDm8 split-Gal4 driver | *yw hsflp/+;*<br>*UAS > CD2, y+>mCD8::GFP/R24F06-p65.AD;*<br>*Rh4-lacZ/DIP-γ$^{MI03222}$Gal4.DBD* |
| *Figure 5—source data 1* | DIP-γ $^{-/-}$: yDm8 split -Gal4 driver | *yw hsflp/+;*<br>*UAS > CD2, y+>mCD8::GFP/R24F06-p65.AD;*<br>*DIP-γ$^{null}$/DIP-γ$^{MI03222}$Gal4.DBD* |
| *Figure 5—source data 1* | dpr11 $^{-/-}$:<br>Pan-Dm8 driver | *yw hsflp/+; UAS > CD2, y+>mCD8::GFP/Rh4LacZ;*<br>*R24F06-Gal4, dpr11$^{null}$/dpr11$^{null}$* |
| *Figure 7—figure supplement 1* | DIP-γ>$^{>GFP}$, DIP-γ- Gal4 > UAS-DIAP$^{myc}$ | *UAS-diap1.myc, DIP-γ$^{MI03222-GFP}$/*<br>*DIP-γ$^{MI03222}$ Gal4* |
| *Figure 8—figure supplement 1A–A'* | | *DIP-γ$^{MI03222-GFP}$/+* |
| *Figure 8—figure supplement 1B–B'* | | WT control |
| *Figure 8—figure supplement 1C* | | *dpr11$^{MI02231-GFP}$/+* |
| *Figure 8—figure supplement 2A,C* | DIP-γ $^{-/+}$ | *DIP-γ$^{MI03222-GFP}$/+* |
| *Figure 8—figure supplement 2B,D* | DIP-γ $^{-/-}$ | *DIP-γ$^{MI03222-GFP}$/DIP-γ$^{null}$* |
| *Figure 9—figure supplement 1A* | ss$^{GOF}$ | *lGMR-ss/+; DIP-γ$^{MI03222-GFP}$* |
| *Figure 9—figure supplement 1B* | ss$^{GOF}$; DIP-γ $^{-/-}$ | *lGMR-ss/+; DIP-γ$^{MI03222-GFP}$/DIP-γ$^{null}$* |
| *Figure 9—figure supplement 1C* | WT | *lGMR-Gal4/+; DIP-γ$^{MI03222-GFP}$/+* |
| *Figure 9—figure supplement 1D* | dpr11$^{GOF}$ | *lGMR-Gal4/UAS-dpr11$^{sh}$; DIP-γ$^{MI03222-GFP}$/+* |
| *Figure 9—figure supplement 1E* | lGMR-Gal4 > UAS-Dve | *lGMR-Gal4/+; DIP-γ$^{MI03222-GFP}$/UAS-dveA-9B2* |
| *Figure 9—figure supplement 1F* | sev $^{-/-}$ | *sev$^{14}$/sev$^{14}$;; R24F06Gal4/DIP-γ$^{MI03222-GFP}$, UAS-mCD8RFP* |
| *Figure 9—figure supplement 1G* | WT | *lGMR Gal4/+* |
| *Figure 9—figure supplement 1H* | lGMR-Gal4 > UAS-dpr11 | *lGMR Gal4/+; UAS-dpr11$^{sh}$* |
| *Figure 9—figure supplement 1I* | dpr11$^{GOF}$ | *lGMR Gal4/Rh4-lacZ; UAS-dpr11$^{sh}$* |
| *Figure 9—figure supplement 1J* | ss$^{GOF}$ | *lGMR-ss/+* |
| *Figure 9—figure supplement 1K* | lGMR-Gal4 > UAS-Dve | *lGMR-Gal4/+; DIP-γ$^{MI03222-GFP}$/UAS-dveA-9B2* |
| *Figure 9—figure supplement 1L–L'* | lGMR-Gal4 > UAS-dpr11 | *lGMR Gal4/+; UAS-dpr11$^{sh}$* |
| *Figure 9—figure supplement 2A–A'* | WT | *dpr11$^{MI02231-GFP}$/+* |
| *Figure 9—figure supplement 2B–B'* | dpr11 $^{-/-}$ | *dpr11$^{MI02231-GFP}$/dpr11$^{null}$* |
| *Figure 9—figure supplement 2C–C"* | dpr11 $^{-/-}$ | *Traffic jam Gal4/UASp DIAP; dpr11$^{MI02231-GFP}$/dpr11$^{null}$* |
| *Figure 9—figure supplement 2D* | dpr11 $^{+/-}$ | *dpr11$^{MI02231-GFP}$/+* |
| *Figure 9—figure supplement 2D* | dpr11 $^{-/-}$ | *dpr11$^{MI02231-GFP}$/dpr11$^{null}$* |
| *Figure 9—figure supplement 2D* | Dm8 DIAP rescue | *Traffic jam Gal4/UASp DIAP; dpr11$^{MI02231-GFP}$/dpr11$^{null}$* |
| *Figure 9—figure supplement 2E* | dpr11 $^{+/-}$ | *Rh4-lexA::p65, lexAop2-brp-short$^{cherry}$/+; dpr11$^{null}$/+* |
| *Figure 9—figure supplement 2E* | dpr11 $^{-/-}$ | *Rh4-lexA::p65, lexAop2-brp-short$^{cherry}$/+; dpr11$^{null}$/dpr11$^{null}$* |

## Immunohistochemistry

Eclosed flies (less than 3 days old) were dissected in phosphate-buffered saline (PBS) and fixed in 4% paraformaldehyde in PBS with 0.2% Triton-X-100 (PBT) for 20 min at room temperature. Brains were washed overnight in PBT, followed by a two-day incubation at 4°C with primary antibody that was diluted in blocking buffer (5% normal goat serum in PBT). Samples were then incubated with secondary antibody (similarly diluted in blocking buffer) for two-days, followed by washing with PBT and PBS and stored in Vectashield.

For single cell flipouts, brains were given heat-shock at 50 hr APF for 10–20 min at 37°C and 1 day eclosed flies were dissected for staining as above.

For cell counts and column analyses, optic lobes were separated from the central brain and mounted top-down. Confocal images were acquired on Zeiss LSM800 or LSM700 microscopes with a 40xobjective. For cell counts and single-cell flipouts, optic lobes were imaged with 1.5 µm and 0.8 µm z-sections, respectively. Single slices or maximum intensity projections were exported with Zen software (Zeiss) for image processing.

## Image processing

Images were processed with Adobe Photoshop.

## Quantitation and statistics

Cell numbers were determined blind to genotype using Zen software to do manual counts. 1-3d old adults were used for counts in adults. For all data (cell soma, columns and Brp overshoots), only one optic lobe per animal was used for counts and statistics. p-value was determined using Student's unpaired t-test from Graphpad Prism. All data reported in graphs are mean + /- standard deviation. yDm8 sprig parameters were measured using Image J software. To obtain measurement of the widest point of a sprig, a slice at which the image was most in focus was selected. The height was measured from the most distal point of the sprig to its most proximal point before the processes of the sprig connected back to the dendritic base. The Brp overshoots were analyzed blind using Image J software, by an individual unrelated to that experiment.

For quantitation of neighboring contacts in all genotypes, well isolated clones were chosen for this analysis. The presence of a single axonal process leading to a soma confirmed that each clone was indeed a single cell. Images were then viewed in Imaris. The identity of the home column was excluded from the yellow vs. pale R7 contacts.

## ExM method

Single-cell flipouts were processed for expansion microscopy after immunohistochemistry (and confirming the presence of flipouts) according to a protocol from Tim Mosca, based on *Mosca et al. (2017)*. Briefly, labeled brains were washed in 50% PBT-PBS mixture before proceeding with steps for anchoring proteins to the sodium acrylate matrix. Gelling chambers were made on coated slides with No.1 coverslips as bridge and brains were embedded in sodium acrylate gel. After incubation for 2 hr at 37°C, the samples were excised from the solidified gel and processed for digestion with proteinase K in six-well plates. Gel fragments were expanded by replacing the digestion buffer with several washes of water. Finally, in preparation for confocal imaging, the gel fragments containing the expanded transparent brains were placed on a 24 mm x 50 mm No. 1.5 coverslip that was coated with poly-l-lysine. Preps were always covered in water to make sure they did not dry out during imaging. Expanded brains were imaged on Zeiss LSM800 with 40x Objective 1.1 NA.

## 3D reconstruction of ExM samples

ExM images were processed for 3D reconstruction using Imaris software (Bitplane). Using surface rendering and fluorescence thresholding, we were able to obtain an accurate 3D structure of a yDm8 in wild-type and *DIP-γ* mutant. Each channel was recreated by adding an additional surface. The threshold for new surface detection was set as needed per image minimizing the addition of surface fragments not present in the original image. Each new surface was segmented in order to isolate the appropriate region of interest. Video recordings were captured at 10 frames per second for a total of 300 frames.

## Acknowledgements

We thank Larry Zipursky, Aljoscha Nern, Takashi Suzuki, and Michael Reiser for fly lines and discussions, Maximilien Courgeon and Claude Desplan for sharing unpublished antibodies and fly lines and for discussions, and Tim Mosca for his ExM protocol. We thank Violana Nesterova for figure preparation, Shuwa Xu and Larry Zipursky for comments on the manuscript, Shuwa Xu for discussions during the course of the work, and Namrata Bali for help with ExM. The *dpr11* RNAi stock was obtained from the Vienna Drosophila Resource Center (VDRC, www.vdrc.at). Stocks obtained from the Bloomington Drosophila Stock Center (NIHP40OD018537) were used in this study. Imaging was performed in the Biological Imaging Facility, with the support of the Caltech Beckman Institute and the Arnold and Mabel Beckman Foundation. Anti-Pros MR1A (mouse 1:4), anti-Elav 7E8A10 (rat, 1:10), anti-Dac 2–3 (mouse 1:50), anti-chaoptin 24B10 (mouse 1:20) were obtained from the Developmental Studies Hybridoma Bank (University of Iowa, IA). This work was supported by NIH grants RO1 EY028116 and R37 NS28182 to KZ, and by the Howard Hughes Medical Institute (S-YT).

## Additional information

### Funding

| Funder | Grant reference number | Author |
| --- | --- | --- |
| National Institutes of Health | R37 NS28182 | Kai Zinn |
| National Institutes of Health | RO1 EY028116 | Kai Zinn |
| Howard Hughes Medical Institute | | Shin-ya Takemura |

The funders had no role in study design, data collection and interpretation, or the decision to submit the work for publication.

### Author contributions

Kaushiki P Menon, Conceptualization, Formal analysis, Supervision, Validation, Investigation, Visualization, Methodology, Writing—original draft, Writing—review and editing; Vivek Kulkarni, Formal analysis, Validation, Investigation, Visualization, Methodology, Writing—review and editing; Shin-ya Takemura, Formal analysis, Validation, Investigation, Visualization, Writing—review and editing; Michael Anaya, Resources, Investigation, Methodology; Kai Zinn, Conceptualization, Supervision, Funding acquisition, Writing—original draft, Project administration, Writing—review and editing

### Author ORCIDs

Kaushiki P Menon https://orcid.org/0000-0003-1039-4704
Shin-ya Takemura http://orcid.org/0000-0003-2400-6426
Kai Zinn https://orcid.org/0000-0002-6706-5605

### Decision letter and Author response

Decision letter https://doi.org/10.7554/eLife.48935.sa1
Author response https://doi.org/10.7554/eLife.48935.sa2

## Additional files

### Data availability

All data generated or analysed during this study are included in the manuscript and supporting files.

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
