## [Decision Letter]

**Acceptance summary:**

This is a very interesting important study that describes a molecular mechanism to selectively assemble and label two developmentally and functionally similar UV detection circuits by expressing interacting IG domain proteins on the neuronal membranes. The study nicely demonstrates the how neuronal circuits maintain functional reciprocity by expressing specific surface molecules. An interesting finding id also that the molecules that generate such synaptic specificity are also trophic factors necessary for cell survival during the construction of the appropriate circuits in the brain. These results have far reaching implications for proper circuit formation in other, more complex situations and organisms.

**Decision letter after peer review:**

Thank you for submitting your article "Interactions with photoreceptors mediated by Dpr11 and DIP-γ control selection and survival of amacrine neurons" for consideration by *eLife*. Your article has been reviewed by three peer reviewers, and the evaluation has been overseen by Hugo Bellen as Reviewing Editor and Utpal Banerjee as the Senior Editor. The reviewers have opted to remain anonymous.

The reviewers have discussed the reviews with one another and the Reviewing Editor has drafted this decision to help you prepare a revised submission.

This article builds on the group's previous report (Carrillo et al., 2015) that Dpr11 is expressed in yR7s, and that its cognate binding partner, DIP-γ is expressed in Dm8 neurons. R7 and Dm8 neurons are synaptic partners. The authors show how DIP-γ is expressed in a subset of Dm8 neurons, whose home column contains a Dpr11 expressing yR7. Using the Takemura et al., 2013, 2015 EM connectome data they describe a specific synaptic connections for pale and yellow R7s proposing two wavelength discrimination circuits, which correlate with Dpr11 expression in yR7 and DIP-γ expression in postsynaptic partners yDm8 and Tm5a. They report subtle abnormalities in the morphology of a specific part of the yDm8 dendritic arbor in Dpr11 and DIP-γ mutants. This region is called the sprig, which is a branch in close contact with the home column yR7. The feature most affected is the maximum diameter of the sprig. The biological meaning of this finding is unclear since although this region is where most R7 out synapses concentrate, in one of the EM reconstructed columns where the sprig is quite reduced output R7 synapses concentrate in the part of the arbor where the base of the sprig is. ExM shows the same yDm8 morphology as EM for wild type cells and the similar sprig defects in as in DIP-γ mutants, however the n of this experiment is not specified. The authors show that Dpr11 and DIP-γ are required, most probably in early pupal development for yDm8 survival. This leads the authors to suggest that yR7 provide trophic support to yDm8. Then the authors show that changes in R7 fate can affect the numbers of yellow and pale Dm8 and force these cells to compete for trophic support. Their results indicate that Dpr11 misexpression in pR7 results in an increase of yDm8 cells and a reduction of pDm8. This seems to be sufficient for pairing Dpr11 expressing R7 cells with DIP-γ yDm8 and mediating their survival. However, the Dpr11 loss of function analysis has not clarified whether Dpr11 is necessary for pairing of yDm8 with yR7.

Essential revisions:

What happens to DIP γ in Dm8 if you convert the yPR7 to pPR7 if you replace Rh3 with Rh4? Do they switch connection specificity or do they die? What happens to *dpr11* expression in Rh4 mutants? The Desplan lab had shown that R8 Rhodopsin selection works through feedback from the receptor, and rhodopsin knock outs lead to expression of the other alternate Rhodopsin in R8. I am not sure if this is tested for R7 Rhodopsins. Still, looking at Rh4 mutants might be better than k/o of a fate determinant which might induce cell death. I guess the question is does Rh4 function regulate Dpr11 expression and whether you can get yPR7 to rewire to pDm8 by specifically k/d Rh4 or replacing Rh3 with Rh4 in yPR7?

The data supporting Drp11-DIP-γ role in promoting yDm8 survival is solid, however the requirement for pairing is unclear. I am wondering if the following experiment would be feasible with existing reagents: inhibit yDm8 death in the Dpr11 mutant background to see whether yDM8s would now occupy or not the vacant yR7 columns.

Since most yDm8 neurons require DIP-γ for survival, we infer that the excess yDm8 neurons in wild-type die because they lack neurotrophic support mediated by Dpr11-DIP-γ interactions. However, we do not understand why yDm8 neurons begin to die earlier in DIP-γ mutants than in wild-type (Figure 7A). Perhaps DIP-γ not engaged by Dpr11 (which would be present in wild-type yDm8) allows cells to survive longer than when DIP-γ is absent. These selection events are independent of synaptogenesis between R7 and Dm8, which occurs only after 60h APF. Can you rescue the lost *dpr11*-DIP γ interactions in mutants with another DIP γ or Dpr11 partner, or a completely different pair like DIP α – dpr10, to see if you can induce synapses between yPR7-Dm8 and whether re-establishing synaptic connections some other means can rescue the death?

The authors do not adequately demonstrate loss of neurotrophic signals in DIP-γ and *dpr11* mutants. They show that there is a cell non-autonomous effect on cell survival but they need to show that expression of neurotrophic genes is lost in these mutants to make this claim more compelling. Alternatively, they need to remove this statement.

---

## [Author Response]

Essential revisions:What happens to DIP γ in Dm8 if you convert the yPR7 to pPR7 if you replace Rh3 with Rh4? Do they switch connection specificity or do they die? What happens to dpr11 expression in Rh4 mutants? The Desplan lab had shown that R8 Rhodopsin selection works through feedback from the receptor, and rhodopsin knock outs lead to expression of the other alternate Rhodopsin in R8. I am not sure if this is tested for R7 Rhodopsins. Still, looking at Rh4 mutants might be better than k/o of a fate determinant which might induce cell death. I guess the question is does Rh4 function regulate Dpr11 expression and whether you can get yPR7 to rewire to pDm8 by specifically k/d Rh4 or replacing Rh3 with Rh4 in yPR7?

The paper the reviewers are referring to is Vasiliauskas et al., Nature 479, 108 (2011). This paper shows that in *rh6* mutants feedback from lack of Rh6 expression causes upregulation of Rh5 (normally expressed only in pale R8s) in yellow R8s. However, this begins to happen only in adults beginning at 3 days after eclosion, and is complete by 14 days. The fate of the yellow R8s is not changed, only the expression of Rh5 (for example, Rh6-lacZ does not turn off). In the paper they show that nothing like this happens in R7s (that is, *rh3* and *rh4* mutations have no effect on rhodopsin expression in the other subtype of R7s). In any case, the phenomenon of upregulation of Rh5 in R8 is clearly unrelated to control of R7 or Dm8 fate by Dpr11 expression, since it occurs only in aging adult flies. Dpr11 turns on in early pupae (by 20 hr. APF), and regulation of Dm8 cell death by Dpr11-DIP-γ interactions is complete by 45 hr. APF (Figure 7). Rh4 expression does not begin until 80 hr. APF (Earl and Britt, Gene Expression Patterns 6, 687 (2006). Therefore, it would make no sense to examine Dpr11 expression in an *rh4* mutant. We have, however, done the reverse experiment, and these results are reported in Figure 8—figure supplement 1. Here we show that ectopic Dpr11 expression in pR7s does not cause them to turn on Rh4-lacZ. Therefore, expressing Dpr11 in pR7s does not change their fates to convert them into yR7s.

The data supporting Drp11-DIP-γ role in promoting yDm8 survival is solid, however the requirement for pairing is unclear. I am wondering if the following experiment would be feasible with existing reagents: inhibit yDm8 death in the Dpr11 mutant background to see whether yDM8s would now occupy or not the vacant yR7 columns.

We have done the suggested experiment, and the results are reported in Figure 8—figure supplement 2. We inhibited cell death by expressing DIAP in Dm8 neurons in a *dpr11* mutant and analyzed home column occupancy with DIP-γ antibody at 40-42h APF. Yellow columns were marked with GFP and Chaoptin in dpr11^Mi>GFP^ background, as in the other panels of Figure 8—figure supplement 2. As the reviewer suggests, we find that yR7s that were unoccupied in the *dpr11* mutant due to death of yDm8s are now occupied when cell death is rescued by DIAP. We did not observe extensive mispairing of yDm8s with pR7s, however, indicating that the rescued yDm8s are not free to innervate pR7s, because pR7s are preferentially innervated by pDm8s. This is consistent with results elsewhere in the paper showing that mistargeting is rare in *DIP-γ* and *dpr11* mutants.

Since most yDm8 neurons require DIP-γ for survival, we infer that the excess yDm8 neurons in wild-type die because they lack neurotrophic support mediated by Dpr11-DIP-γ interactions. However, we do not understand why yDm8 neurons begin to die earlier in DIP-γ mutants than in wild-type (Figure 7A). Perhaps DIP-γ not engaged by Dpr11 (which would be present in wild-type yDm8) allows cells to survive longer than when DIP-γ is absent. These selection events are independent of synaptogenesis between R7 and Dm8, which occurs only after 60h APF. Can you rescue the lost dpr11-DIP γ interactions in mutants with another DIP γ or Dpr11 partner, or a completely different pair like DIP α – dpr10, to see if you can induce synapses between yPR7-Dm8 and whether re-establishing synaptic connections some other means can rescue the death?

The suggested replacement experiment would be of interest. However, replacement of DIP-γ and Dpr11 with DIP-α and Dpr10 would require introducing several transgenes, mutations, GAL4, and lexA drivers into the same line. If such a line could even be built, it would require many months to produce. We have, however, addressed this point by ectopically expressing DIP-α and DIP-γ in all PRs and asked if these manipulations produces cell death of yDm8 neurons. We observed that PR expression of DIP-γ results in extensive yDm8 cell death, presumably because Dpr11 is sequestered by *cis* interactions with DIP-γ in yR7s and thus cannot engage with DIP-γ on yDm8s. However, PR expression of DIP-α, which cannot bind to Dpr11 or DIP-γ, has no effect on yDm8 survival, These results are reported in Figure 6 of the revised version of the manuscript. We also investigated whether doing the reverse manipulation by expressing Dpr11 in yDm8s would cause cell death, but observed that this had no effect.

The authors do not adequately demonstrate loss of neurotrophic signals in DIP-γ and dpr11 mutants. They show that there is a cell non-autonomous effect on cell survival but they need to show that expression of neurotrophic genes is lost in these mutants to make this claim more compelling. Alternatively, they need to remove this statement.

We agree that *dpr11* and *DIP-γ* mutations have not been demonstrated to have neurotrophic effects, if this is defined by loss of expression of genes required for neurotrophic signaling. There are several different neurotrophic pathways that have been identified in *Drosophila*, and it is unknown which ones might be involved in regulation of Dm8 cell death. We have therefore removed all references to neurotrophic signaling in the paper. We now just state that Dpr11-DIP-γ interactions allow yDm8s to survive.

We also added an experiment showing that yDm8s express a cell death marker, Apoliner, at 25 hr. APF in *DIP-γ* mutants, and that many fewer yDm8s express Apoliner in wild-type. At this time, cell loss is already occurring in *DIP-γ* mutants but not in wild-type. These results have been added to Figure 7.